# The hepatic AMPK-TET1-SIRT1 axis regulates glucose homeostasis

Chunbo Zhang[1,2†], Tianyu Zhong[1†], Yuanyuan Li[3†], Xianfeng Li[3†], Xiaopeng Yuan[2], Linlin Liu[4], Weilin Wu[1], Jing Wu[1], Ye Wu[2], Rui Liang[5], Xinhua Xie[1], Chuanchuan Kang[4], Yuwen Liu[1], Zhonghong Lai[1], Jianbo Xiao[1], Zhixian Tang[4], Riqun Jin[4], Yan Wang[3], Yongwei Xiao[1], Jin Zhang[6], Jian Li[5]*, Qian Liu[1]*, Zhongsheng Sun[3]*, Jianing Zhong[1]*

[1]Key Laboratory of Prevention and Treatment of Cardiovascular and Cerebrovascular Diseases of Ministry of Education, Gannan Medical University, Ganzhou, China; [2]School of Pharmacy, Nanchang University, Nanchang, China; [3]Beijing Institutes of Life Science, Chinese Academy of Sciences, Beijing, China; [4]Precision Medicine Center, First Affiliated Hospital of Gannan Medical University, Ganzhou, China; [5]State Key Laboratory of Cellular Stress Biology, Innovation Center for Cell Biology, School of Life Sciences, Xiamen University, Xiamen, China; [6]School of Basic Medical Sciences, Nanchang University, Nanchang, China

*For correspondence:
jianli_204@xmu.edu.cn (JL);
liuqiangmu2017@126.com (QL);
sunzs@biols.ac.cn (ZS);
zhongning_003@163.com (JZ)

†These authors contributed equally to this work

**Abstract** Ten-eleven translocation methylcytosine dioxygenase 1 (TET1) is involved in multiple biological functions in cell development, differentiation, and transcriptional regulation. *Tet1* deficient mice display the defects of murine glucose metabolism. However, the role of TET1 in metabolic homeostasis keeps unknown. Here, our finding demonstrates that hepatic TET1 physically interacts with silent information regulator T1 (SIRT1) *via* its C-terminal and activates its deacetylase activity, further regulating the acetylation-dependent cellular translocalization of transcriptional factors PGC-1α and FOXO1, resulting in the activation of hepatic gluconeogenic gene expression that includes *PPARGC1A*, *G6PC*, and *SLC2A4*. Importantly, the hepatic gluconeogenic gene activation program induced by fasting is inhibited in *Tet1* heterozygous mice livers. The adenosine 5'-monophosphate-activated protein kinase (AMPK) activators metformin or AICAR—two compounds that mimic fasting—elevate hepatic gluconeogenic gene expression dependent on in turn activation of the AMPK-TET1-SIRT1 axis. Collectively, our study identifies TET1 as a SIRT1 coactivator and demonstrates that the AMPK-TET1-SIRT1 axis represents a potential mechanism or therapeutic target for glucose metabolism or metabolic diseases.

## Editor's evaluation

The authors identify the DNA demethylase Tet1 as being a critical regulator of hepatic carbohydrate metabolism. Moreover, the authors discover these effects are mediated through a novel non-canonical function of Tet1 independent of altering DNA hydroxymethylation.

## Introduction

Ten-eleven translocation methylcytosine dioxygenase 1 (TET1), as a member of the Tet family, has the capacity to convert 5-methylcytosine to 5-hydroxymethylcytosine (5hmC) in a 2-oxoglutarate- and Fe(II)-dependent manner and is involved in DNA demethylation (*Koivunen and Laukka, 2018*). Many studies suggest that *Tet1* heterozygous mice exhibit variable phenotypes, including placental, fetal, and postnatal growth defects, as well as early embryonic lethality, partly caused by imprinted gene

dysregulation (*Yamaguchi et al., 2013*). Tet2 has also recently been associated with the glucose-AMPK-TET2-5hmC axis signaling pathway, linking the level of extracellular glucose to the dynamic epigenetic regulation of 5hmC with implications in diabetes and cancer (*Wu et al., 2018*). Although there are many studies on the DNA demethylation function of TET1, few studies have specifically evaluated the regulation and function of TET1 in glucose homeostasis.

Silent information regulator T1 (SIRT1) is an enzyme that mediates nicotinamide adenine dinucleotide NAD$^+$-dependent deacetylation of target substrates. Due to its ability to modify many transcriptional factors and co-factors involved in glucose homeostasis, hepatic SIRT1 is referred to as a key metabolic regulator (*Chang and Guarente, 2014*; *Yu et al., 2018*). In response to fasting or calorie restriction, hepatic SIRT1-mediated deacetylation and activation of PGC-1α improve glucose homeostasis (*Rodgers et al., 2005*). Additionally, the protein PGC-1α together with FOXO1—a crucial regulatory transcriptional factor in various metabolic processes—play critical roles in the development of obesity, insulin resistance, and type 2 diabetes (*Asher and Schibler, 2011*; *Housley et al., 2009*), supporting the notion that the hepatic SIRT1-PGC-1α/FOXO1 axis is part of the classic metabolic sensing network.

Mechanically, the deacetylation of PGC-1α or FOXO1 modulated by SIRT1 is tightly linked with enhanced transcriptional activation in gluconeogenesis and glycolysis, resulting in increased hepatic glucose production and further contributing to the development of type 2 diabetes (*Rodgers et al., 2005*; *Puigserver et al., 1998*; *Feige and Auwerx, 2007*). In response to fasting, adenosine 5'-monophosphate (AMP)-activated protein kinase (AMPK)—another metabolic sensor—can be activated by phosphorylation of its Ser/Thr 172 residue, and then phosphorylated AMPK can directly modulate SIRT1 activation regarding PGC-1α deacetylation and further activate the glucose metabolism pathway (*Cantó et al., 2009*; *Cantó and Auwerx, 2009*). Recent insight from different in vivo transgenic models suggests that AMPK, SIRT1, and PGC-1α/FOXO1 act as an orchestrated network to improve metabolic fitness.

Here, we directly link TET1 activation to the induction of gene expression in glucose metabolism and downstream metabolic programs, thereby defining a critical regulatory axis among AMPK, SIRT1, and PGC-1α/FOXO1. Mechanically, TET1 has a critical effect in mice on glucose metabolism by directly interacting with SIRT1 and increasing the deacetylase activity of SIRT1 on PGC-1α/FOXO1. The SIRT1-mediated acetylation-dependent translocation PGC-1α/FOXO1 enhances gene expression in gluconeogenesis signaling pathways, ultimately leading to regulation of hepatic glucose metabolism. Importantly, *Tet1* heterozygous mice exhibit the defects of blood glucose homeostasis in response to fasting or metformin injections. In conclusion, our work identifies Tet1 as an important metabolic regulator in liver cells and proposes a potential pathway that plays a critical role in regulating glucose homeostasis.

## Results

### Tet1 deficiency disrupts murine glucose metabolism

Since *Tet1* homozygous mice (*Tet1$^{-/-}$*) exhibit embryonic lethality (*Yamaguchi et al., 2012*; *Dawlaty et al., 2011*), we used *Tet1* heterozygous mice (*Tet1$^{+/-}$*) and verified its transcriptional expression of Tet1 (*Figure 1—figure supplement 1A*). *Tet1$^{+/-}$* and wild-type (WT) mice with Chow diet feeding showed no significant differences in body weight, food intake, and blood glucose levels (*Figure 1—figure supplement 1B and C*). The blood insulin level was slightly lower in *Tet1$^{+/-}$* mice compared with WT mice but was not significantly different (*Figure 1—figure supplement 1D*). There was no significant differences between *Tet1$^{+/-}$* mice and WT mice by H&E staining in epididymal white adipose tissue, inguinal white adipose tissues, and liver (*Figure 1—figure supplement 1E and F*). To investigate Tet1 function in glucose metabolism, we performed glucose tolerance test (GTT), pyruvate tolerance test (PTT), and insulin tolerance test (ITT). The glucose level in *Tet1$^{+/-}$* mice was more readily responsive to glucose, pyruvate, and insulin injections than WT mice (*Figure 1A, B and C*), indicating that Tet1 deficiency improves both glucose and insulin tolerance and exhibits a lower gluconeogenesis rate compared to WT mice. These data together indicate that Tet1 deficiency affects glucose metabolism.

To further confirm the effect of Tet1 on glucose metabolism, mice were fed with a high-fat diet (HFD) for 12 weeks. The body weight and food intake were similar between *Tet1$^{+/-}$* and WT mice (*Figure 1—figure supplement 2A,B*). The insulin level was not significantly changed between HFD-fed

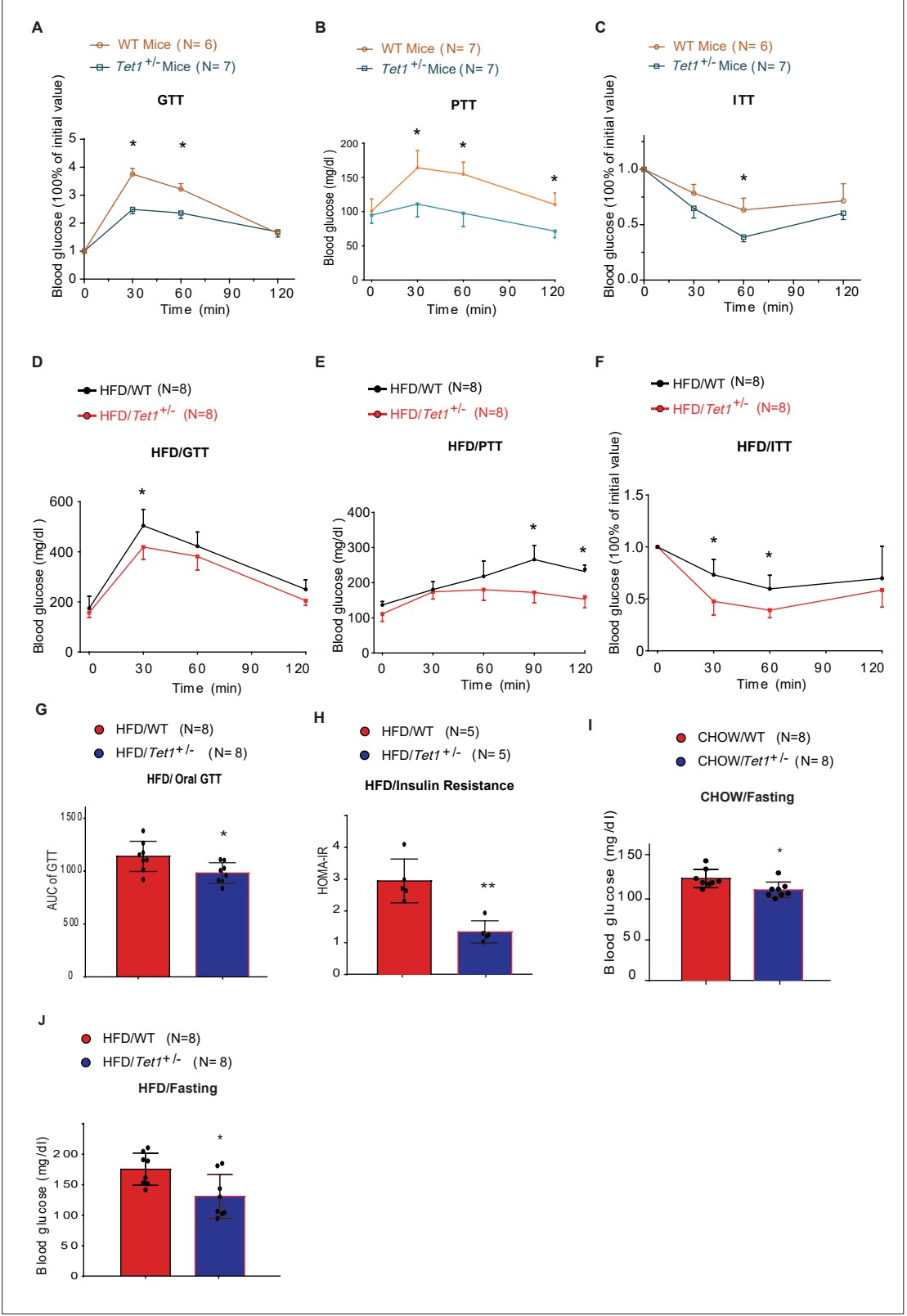

**Figure 1.** Tet1 deficiency leads to murine glucose metabolism defect. (**A**) Glucose tolerance test (GTT) assays in *Tet1*[+/-] male mice and wild-type (WT) male mice (*p < 0.05). (**B**) Pyruvate tolerance test (PTT) assays in *Tet1*[+/-] male mice and WT male mice (*p < 0.05). (**C**) Insulin tolerance test (ITT) assays in *Tet1*[+/-] male mice and WT male mice (*p < 0.05). (**D**) GTT assays in HFD-fed *Tet1*[+/-] male mice and WT male mice (*p < 0.05). (**E**) PTT assays in high-fat diet (HFD)-fed *Tet1*[+/-] male mice and WT male mice (*p < 0.05). (**F**) ITT assays in HFD-fed *Tet1*[+/-] male mice and WT male mice (*p < 0.05). (**G**) Area

*Figure 1 continued on next page*

Figure 1 continued

under the curve (AUC) of oral GTT assays in HFD-fed *Tet1*[+/-] male mice and WT male mice (*p < 0.05). (**H**) Homeostatic Model Assessment for Insulin Resistance (HOMA-IR) assay in HFD-fed *Tet1*[+/-] male mice and WT male mice (**p < 0.01). (**I**) The blood glucose levels in *Tet1*[+/-] male mice and WT male mice with fasting for 24  hr (*p < 0.05). (**J**) The blood glucose levels in HFD-fed *Tet1*[+/-] male mice and WT male mice with fasting for 24  hr (*p < 0.05).

The online version of this article includes the following figure supplement(s) for figure 1:

**Figure supplement 1.** The body weight, food intake, and serum insulin of *Tet1*[+/-] mice has no alternation compared with wild-type (WT) mice.

**Figure supplement 2.** The body weight of high-fat diet (HFD)-fed *Tet1*[+/-] mice has no alternation compared with HFD-fed wild-type (WT) mice.

**Figure supplement 3.** The detection of oral glucose tolerance, insulin resistance, and blood glucose in mice.

*Tet1*[+/-] and WT mice (**Figure 1—figure supplement 2C**). Further, using GTT, ITT, and PTT assays, we found lower blood glucose concentrations in HFD-fed *Tet1*[+/-] male mice treated with glucose, insulin, and pyruvate compared to WT mice (**Figure 1D–F**). The area under the curve (AUC) of GTT confirmed the improved glucose tolerance in HFD-fed *Tet1*[+/-] mice (**Figure 1G**), but not in Chow-fed mice (**Figure 1—figure supplement 3A**). The Homeostatic Model Assessment for Insulin Resistance (HOMA-IR) assay showed that blood glucose levels in HFD-fed *Tet1*[+/-] mice more readily respond to insulin injection than WT mice (**Figure 1H**), but not in Chow-fed mice (**Figure 1—figure supplement 3B**), suggesting a beneficial effect of Tet1 deficiency in the development of insulin resistance in HFD-fed mice. These results demonstrate that Tet1 deficiency improves glucose tolerance and insulin sensitivity in the HFD feeding condition.

Interestingly, although blood glucose in Chow-fed *Tet1*[+/-] mice was not significantly different than WT mice (**Figure 1—figure supplement 3C**), blood glucose levels were significantly lower in Chow-fed *Tet1*[+/-] mice compared with WT mice after 12 hr of fasting (**Figure 1I**). Similar results were found in the HFD-fed *Tet1*[+/-] mice compared with the HFD-fed WT mice after 12 hr of fasting (**Figure 1J**), but not in non-fasting condition (**Figure 1—figure supplement 3D**). In conclusion, our results demonstrate that Tet1 deficiency suppresses gluconeogenesis as well as improves glucose tolerance and insulin resistance in mice.

## Tet1 deficiency regulates transcriptional expression of glucose metabolism

To identify the mechanism by which Tet1 is involved in glucose metabolism, we detected the transcriptional expression of key genes in the glycolysis, gluconeogenesis, and glutathione metabolism pathways by Real-time PCR in TET1 knockdown HepG2 cells and control cells, including *ALDOA*, *ALDH1A3*, *SLC2A4*, *G6PC*, *PPARGC1A*, *GSR1*, *GCLC*, and *RRM2* (**Figure 2A**). The results showed that TET1 knockdown can lead to the increase of the selected genes in glycolysis and decrease of the genes in gluconeogenesis. We further verified the transcriptional expression (**Figure 2—figure supplement 1A**) and protein level (**Figure 2—figure supplement 1B**) of *PPARGC1A*, *G6PC*, and *SLC2A4* genes in TET1 knockdown LO2 cells using two shRNA, respectively, suggesting TET1 can play important roles in the expression regulation of *PPARGC1A*, *G6PC*, and *SLC2A4* genes. Meanwhile, we also determined the protein expression of *Ppargc1a* and *Slc2a4* by immunofluorescence in different tissues, including liver, kidney, and skeletal muscle (**Figure 2B**). Statistical analysis showed that the protein level of them was significantly decreased in Tet1 knockout mice (**Figure 2C**).

## The TET1 C-terminal interacts with SIRT to increase deacetylase activity

TET1 protein contains a conserved double-stranded β-helix domain, a cysteine-rich domain, and binding sites for the co-factors Fe2+ and 2-oxoglutarate that together form the core catalytic region in the C-terminus. We constructed three fragment plasmids of TET1 based on the conserved domain, described as Flag-FL1, Flag-FL2, and Flag-FL3 (**Figure 3A**). To screen the FL3-interacting histone modifiers, we performed mass spectrometry on the immunoprecipitation mixture extracted from overexpressing Flag-FL3 cells, which indicated that the C-terminal of TET1 interacts with SIRT1 (**Figure 3B**). We also detected the co-existence of SIRT1 in the TET1-IP product by Western blot, validating the TET1-SIRT1 protein-protein interaction (**Figure 3C**). The IP data further confirmed that only Flag-FL3 can specifically interact with SIRT1 and SIN3A (a co-factor of SIRT1), (**Figure 3D**) but not other deacetylases, including HDAC1, SIRT2, and SIRT6 (**Figure 3—figure supplement 1A**). Furthermore, *Escherichia coli* purified SIRT1 and SIN3A physically interact with TET1-FL3 by His-pull-down assays

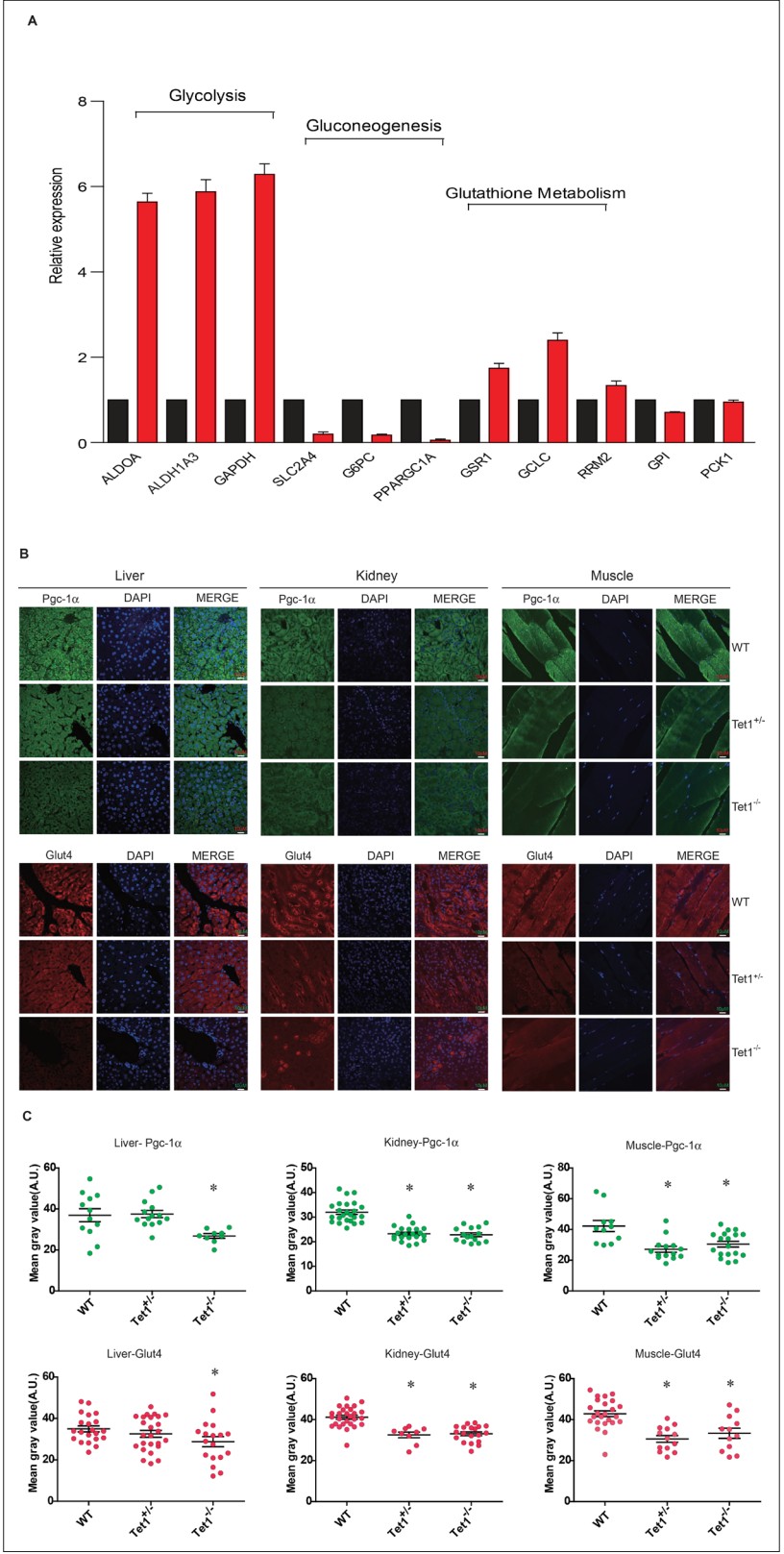

**Figure 2.** Ten-eleven trans-location methylcytosine dioxygenase 1 (Tet1) regulates gluconeogenesis-related transcription expression. (**A**) Transcriptional expression of selected genes in glycolysis, gluconeogenesis, and glutathione metabolism signaling pathways in TET knockdown cells using siRNA. (**B and C**) Representative immunostaining and quantification of signal intensity of liver, kidney, and skeletal muscle from wild-type (WT) and

*Figure 2 continued on next page*

*Figure 2 continued*

transgenic mice using Pgc-1α and Glut4 antibodies. The quantification of the immunofluorescent images was employed via ImageJ software (*p < 0.05).

The online version of this article includes the following source data and figure supplement(s) for figure 2:

**Figure supplement 1.** The transcriptional regulation of *PPARGC1A*, *G6PC*, and *SLC2A4* decreased in ten-eleven translocation methylcytosine dioxygenase 1 (TET1) shRNA knockdown LO2 cells.

**Figure supplement 1—source data 1.** Supplementary Original Data for *Figure 2—figure supplement 1B*.

in vitro assays (*Figure 3E*). Notably, TET1 depletion led to increased levels of the SIRT1-specifically controlled 382 and 120 acetylation sites on p53, which was inhibited by SIRT1 inhibitor EX527 treatment in vivo (*Figure 3F*), suggesting that TET1 regulates SIRT1 deacetylase activity toward downstream substrates. We further purified His-FL3 and then examined SIRT1 deacetylase activity by deacetylase reaction assay in vitro. The SIRT1 deacetylase activity increased following His-FL3 incubation (*Figure 3G*). These results suggest that the C-terminal of TET1 is required for SIRT1 deacetylase activity.

## TET1 regulates PGC-1α and FOXO1 translocation contingent on SIRT-mediated acetylation

Since FOXO1 and PGC-1α are essential transcriptional regulators of gluconeogenesis and regulated by SIRT1 deacetylase activity, we hypothesized that TET1 could play a role in the regulation of PGC-1α and FOXO1. Treatment with EX527—a highly selective and potent inhibitor against SIRT1—followed by immunofluorescence indicates that both PGC-1α and FOXO1 translocate from the nucleus to the cytoplasm (*Figure 4A*), suggesting that the localization of both PGC-1α and FOXO1 were dependent on SIRT1-mediated deacetylation. Also, chromatin fractionation assay performed in EX527-treated cells showed that PGC-1α and FOXO1 were increased in the cytoplasm fraction and decreased in the chromatin fraction (*Figure 4B*). The translocation of both PGC-1α and FOXO1 from nucleus to cytoplasm was similar in TET1-depleted cells as compared to EX527-treated cells (*Figure 4C and D*). Importantly, we further found by immunoprecipitation using an acetyl-specific antibody that PGC-1α and FOXO1 acetylation levels were increased after TET1 depletion, indicating decreased SIRT1 activity (*Figure 4E*). Subsequently, we found that TET1 depletion blocked the increased expression of *PPARGC1A*, *G6PC*, and *SLC2A4* in PGC-1α- or FOXO1-overexpressing cells (*Figure 4F and G*). Taken together, these results suggest that TET1 promotes the deacetylation-dependent translocalization of both PGC-1α and FOXO1 from cytoplasm by SIRT1, thereby down-regulating the transcriptional expression of downstream genes involved in glucose metabolism.

## The TET1-SIRT1 axis regulates pyruvate levels and glycolysis

Gluconeogenesis, which is essentially the reverse process of glycolysis, presented the breakdown of glucose up to formation of pyruvate (*Ma et al., 2013*; *Kinoshita and Kawamori, 2002*). We hypothesized that TET1 inhibition of gluconeogenesis through transcriptional regulation activates glycolysis. The cellular pyruvate level increased in *Tet1*[+/-] mice liver cells and TET1 knockdown in HepG2 cells but decreased upon SIRT1 activator resveratrol (RSV) treatment (*Figure 5A–C*). RSV treatment blocked the increase of cellular pyruvate in TET1 knockdown cells, whereas TET1 knockdown partly rescued the decrease of cellular pyruvate in RSV-treated cells (*Figure 5D and E*). In addition, the decreased glycolysis observed in both the *Tet1*[+/-] mice liver (*Figure 5F*) and TET1-depleted hepatic cells was rescued via RSV treatment (*Figure 5G and H*). These results suggest that TET1 is involved in glucose metabolism via regulating SIRT1. The repression of the transcriptional expression of *G6PC*, *PPARGC1A*, and *SLC2A4* was dosage-dependent on EX527 treatment (*Figure 5—figure supplement 1A*), whereas the activation of the transcriptional expression of these three genes was dosage dependent on RSV treatment (*Figure 5—figure supplement 1B*), suggesting that these genes are strictly regulated by SIRT1 deacetylase activity. RSV and SIRT1-specific activator SRT2104 can increase the protein expression of *G6PC*, *PPARGC1A*, and *SLC2A4* in cells (*Figure 5—figure supplement 1C*). Interestingly, RSV or SRT2104 treatment rescued the decreased gluconeogenic gene expression in TET1-depleted cells (*Figure 5I and J*). TET1 depletion blocked the increase of gluconeogenic gene expression—including *G6PC*, *PPARGC1A*, and *SLC2A4*—in RSV-treated cells (*Figure 5—figure supplement 1D*) or

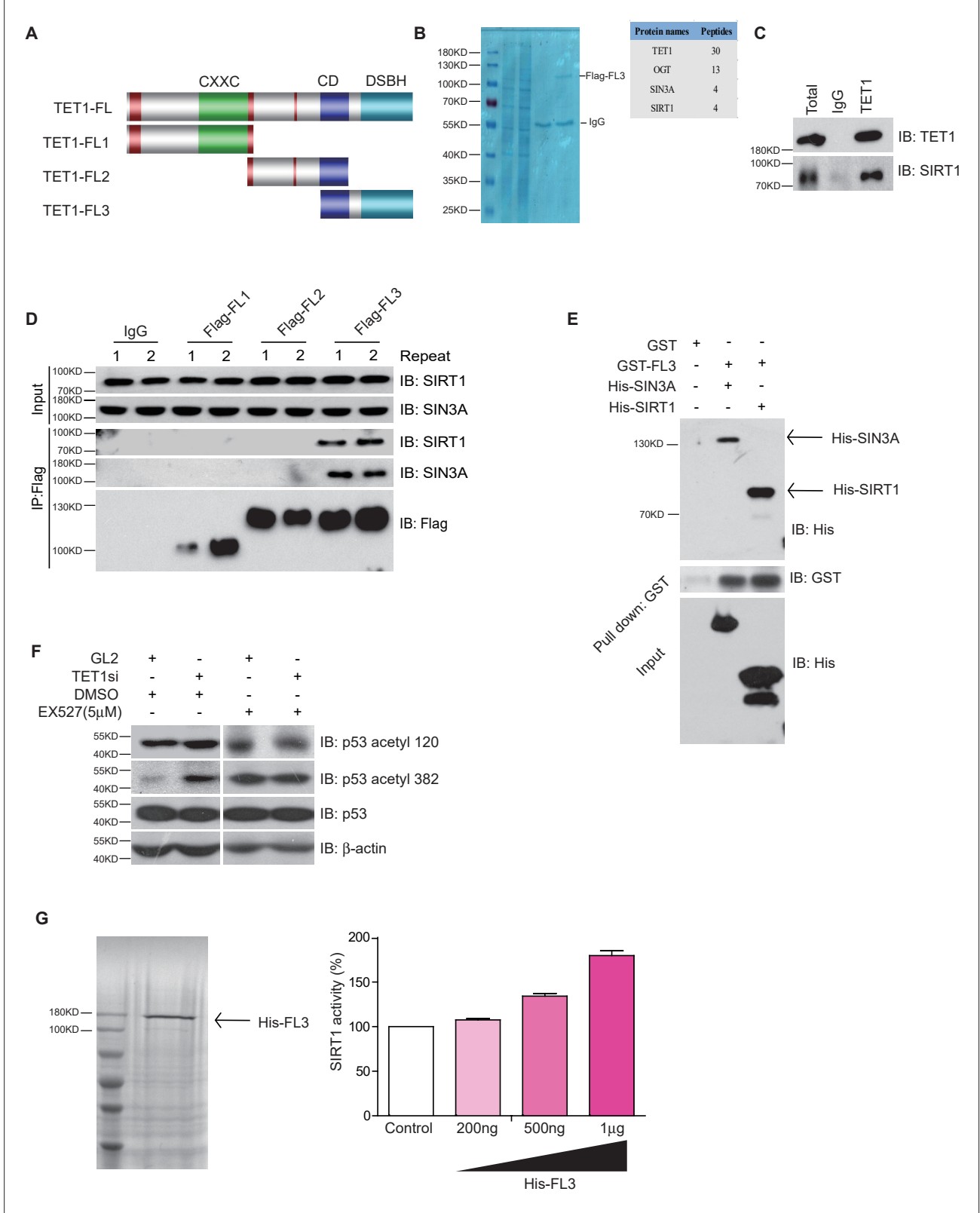

**Figure 3.** Ten-eleven translocation methylcytosine dioxygenase 1 (TET1) interacts with silent information regulator T1 (SIRT1) via its C-terminal to regulate deacetylase activity. (**A**) Schematic representation of TET1 fragments (FL) including Flag-FL1, Flag-FL2, and Flag-FL3 (CXXC: binding CpG islands; CD: cysteine-rich domain; DSBH: double-stranded β-helix). (**B**) Mass spectrum assays were performed for the immunoprecipitation (IP) samples extracted from HepG2 cells using TET1 or SIRT1 antibodies. The overlapped area shown in Venn map represents the interaction proteins

*Figure 3 continued on next page*

*Figure 3 continued*

immunoprecipitated by TET1 or SIRT1. (**C**) TET1 shown interacting with SIRT1 by immunoprecipitation TET1 antibody. (**D**) TET1-FL3 shown interacting with SIRT1 and SIN3A using protein IP in TET1 fragments overexpressing cells. SIN3A is a co-factor of SIRT1 treated as a positive control. Each sample was performed in duplicate. (**E**) GST and GST-FL3 were expressed in BL21 cells and purified following pGEX-GST-vector's manual. His-SIRT1 and His-SIN3A were also expressed in BL21 and purified. Pull-down assays were performed using a GST-tag antibody. (**F**) Western blot analysis of the extracts in TET1 knockdown HepG2 cells and EX527-treated HepG2 cells using specific antibodies as indicated. (**G**) The deacetylase activity of SIRT1 was determined by incubating with 200 ng, 500 ng, and 1 µg His-FL3 for 1 hr. His-FL3 was expressed in BL21 cells and purified following pTrc-His-vector's manual.

The online version of this article includes the following figure supplement(s) for figure 3:

**Figure supplement 1.** Ten-eleven translocation methylcytosine dioxygenase 1 (TET1) cannot interact with HDAC1, silent information regulator T2 (SIRT2), and SIRT6.

SIRT2104-treated cells (*Figure 5—figure supplement 1E*). TET1 depletion rescued glycolysis-related gene expression in RSV-treated cells, including increases in the expression of *ALDOA* and *ALDH1A3* (*Figure 5—figure supplement 2*). In mice, RSV treatment could effectively rescue the decrease of glucose level (*Figure 5—figure supplement 3A and B*), and the defects in glucose and insulin tolerance, and gluconeogenesis of *Tet1*[+/-] mice by GTT, ITT, and PTT assays (*Figure 5—figure supplement 3C and D*). These results suggest that TET1 is involved in the transcriptional regulation of glucose metabolism by regulating SIRT1 deacetylase activity and thus reveal the requirement for TET1-mediated SIRT1 activation of glucose metabolism.

## TET1 regulates glucose metabolism dependent on AMPK activity

Both metformin and fasting activate the AMPK signaling pathway (*Lee et al., 2011*; *Suwa et al., 2006*; *Bujak et al., 2015*; *Kajita et al., 2008*). And AMPK is activated by its phosphorylation on 172 residue (*Cantó et al., 2009*; *Cantó and Auwerx, 2009*). To determine the regulatory relationship between TET1 and AMPK, we performed chromatin fractionation in TET1-depleted hepatic cells. The results show that both AMPK and activated p-AMPK (AMPK-p172) were not changed in the cytoplasmic fraction, suggesting that TET1 depletion could not affect the activation of AMPK (*Figure 6A*). Additionally, SIRT1 inhibition also does not affect the phosphorylation, localization, or protein levels of AMPK (*Figure 6—figure supplement 1*), suggesting that TET1 and SIRT1 are the downstream mediators in the AMPK-dependent signaling pathway. Furthermore, the results show that 5-aminoimidazole-4-carboxamide ribonucleoside (AICAR)—a specific AMPK activator—can rescue the glycolysis defect from TET1 knockdown in HepG2 cells (*Figure 6B*). These results indicate that TET1/SIRT1 acts downstream of AMPK to regulate gene expression.

To further determine the role TET1 plays in the AMPK-dependent transcriptional regulation of glucose metabolism, we examined whether higher concentrations of AICAR and metformin could activate *G6PC*, *PPARGC1A*, and *SLC2A4* gene expression (*Figure 6C and D*). These results indicate that TET1 depletion blocks the activation of these three genes following AICAR or metformin treatment in HepG2 cells (*Figure 6E*). Since glucagon is a sensitive and key regulator of glucose metabolism by activating gluconeogenesis and repressing glycolysis, we also examined how glucagon affected *G6PC*, *PPARGC1A*, and *SLC2A4* gene expression. We found that the transcriptional expression of *G6PC*, *PPARGC1A*, and *SLC2A4* is dependent on glucagon treatment in HepG2 cells (*Figure 6F*). Importantly, TET1 depletion blocks the increase of the expression of these genes in glucagon-treated HepG2 cells (*Figure 6G*). These results indicate that TET1 regulates the transcriptional expression of glucose metabolism dependent on AMPK activation.

To determine the effects of transcriptional expression in glucose metabolism in vivo, we fed the mice with metformin (*Figure 7A*) followed by measures for blood glucose levels and the metformin tolerance test (MTT). Metformin did not affect the body weight in WT mice or *Tet1*[+/-] mice (*Figure 7—figure supplement 1A and B*), but Tet1 deficiency prevented the decrease of blood glucose levels (*Figure 7B*). To further determine metformin-induced activation of these genes in *Tet1*[+/-] mice and WT mice, we found that Tet1 deficiency blocked the transcriptional activation of these genes, including *G6pc* (*Figure 7C*), *Ppargc1a* (*Figure 7D*), and *Slc2a4* (*Figure 7E*), which was induced via metformin treatment. Also, we found similar results that Tet1 deficiency blocked the transcriptional activation of these same genes in the fasting condition (*Figure 7F–H*). These results suggest that Tet1 regulates

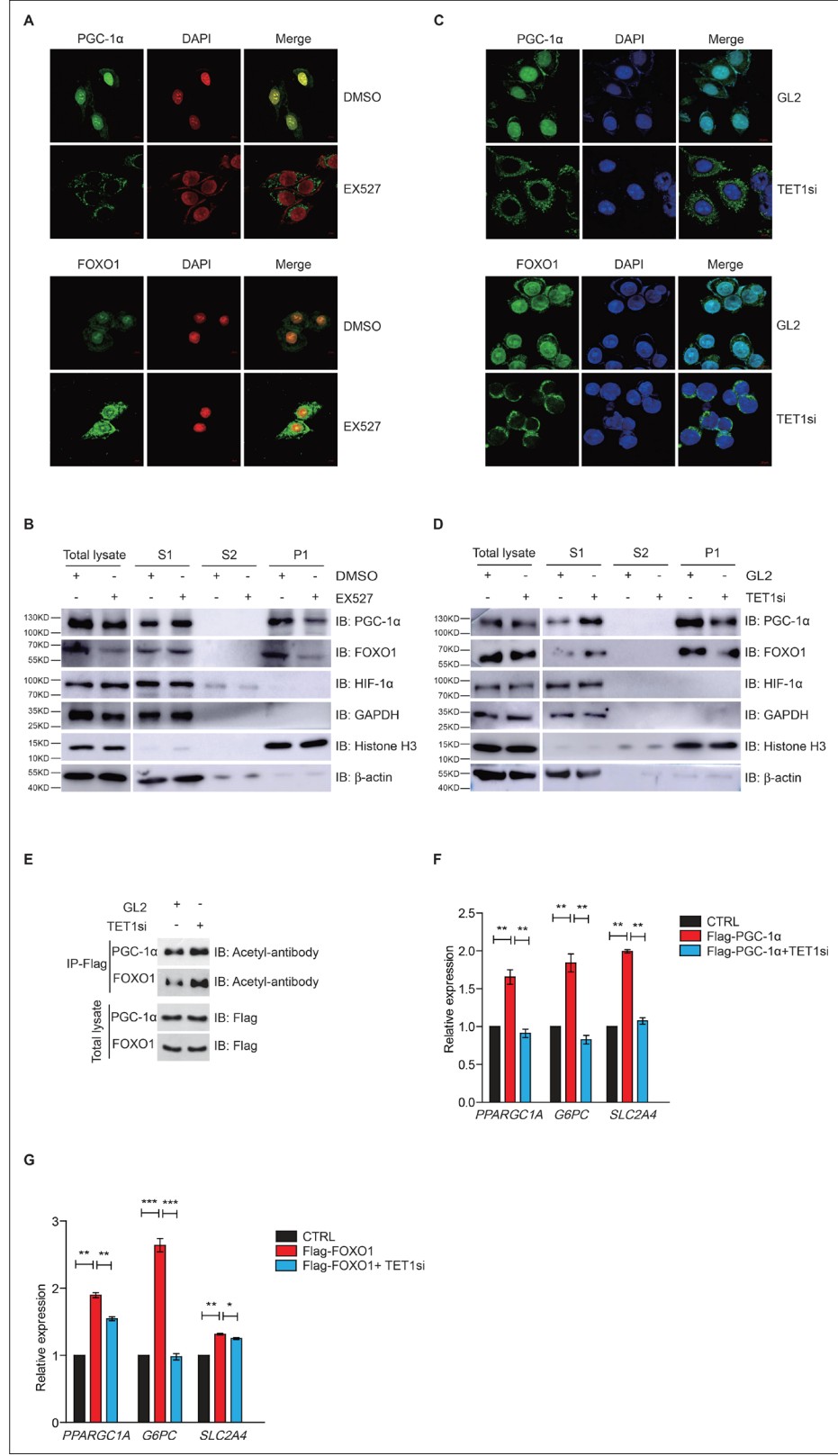

**Figure 4.** Ten-eleven translocation methylcytosine dioxygenase 1 (TET1)-dependent silent information regulator T1 (SIRT1) activity regulates PGC-1α and FOXO1 translocalization. (**A**) Immunofluorescence staining of the translocalization of PGC-1α or FOXO1 in EX527-treated HepG2 cells. Nuclei were stained with DAPI. (**B**) Western blot analysis of the fractions in EX527-treated HepG2 cells using antibodies. S1, S2, and P1 mean the proteins in

*Figure 4 continued on next page*

*Figure 4 continued*

the cytoplasm, nucleus soluble proteins, and chromatin binding proteins, respectively. (**C**) Immunofluorescence staining of the translocalization of PGC-1α or FOXO1 in TET1-depleted or negative control (GL2) HepG2 cells. Nuclei were stained with DAPI. (**D**) Western blot analysis of the fractions in TET1 knockdown HepG2 cells using antibodies. (**E**) Western blot analysis of protein immunoprecipitations in Flag- PGC-1α or Flag-FOXO1 overexpressing cells using acetyl antibodies. (**F**) qRT-qPCR analysis of *PPARGC1A*, *G6PC*, and *SLC2A4* mRNA levels in PGC-1α overexpressed HepG2 cells, with or without TET1 knockdown (**p < 0.01). (**G**) qRT-qPCR analysis of *PPARGC1A*, *G6PC*, and *SLC2A4* mRNA levels in FOXO1 overexpressed HepG2 cells, with or without TET1 knockdown (*p < 0.05, **p < 0.01, ***p < 0.001).

the transcriptional expression of important genes related to glucose metabolism dependent on both metformin and fasting activated signaling pathways in vivo.

## Discussion

In the present study, we demonstrate that TET1 plays a critical role in the AMPK-SIRT1-PGC-1α/FOXO1 signaling pathway for hepatic glucose metabolism. TET1 forms a complex with SIRT1 and regulates its deacetylase activity to deacetylate PGC-1α and FOXO1, maintaining their nuclear localization and thereby activating the transcriptional expression of glucose metabolism-related genes in hepatocytes. Importantly, TET1 deficiency prevents the activation of the transcriptional expression of *G6PC*, *PPARGC1A*, and *SLC2A4* in response to metformin treatment or fasting dependent on AMPK signaling pathway, thus affecting glucose homeostasis in mice. Our study not only uncovers a novel role for TET1 in the transcriptional regulation of hepatic glucose metabolism genes but also reveals a potential molecular mechanism for understanding the roles of TET1 in hepatic glucose metabolism that is dependent on AMPK activation (*Figure 8*).

When blood glucose levels are low, due to fasting or caloric restriction, hepatic metabolism immediately shifts to glycogen breakdown and then gluconeogenesis to ensure glucose supply and ketone body production to bridge energy deficits. *Tet1* heterozygous mice have no alternation in body weight but exhibit a defect in the glucose homeostasis during fasting, similar to the phenotype in SIRT1 knockout mice (*Erion et al., 2009*). It has been well established that the SIRT1-PGC-1α/FOXO1 axis is a core signaling pathway for transcriptional regulation in hepatic glucose metabolism. Fasting or caloric restriction can activate the SIRT1-PGC-1α/FOXO1 axis, in which SIRT1 triggers the next gluconeogenesis by deacetylating HIF-1α, PGC-1α, or FOXO1 to generate glucose output. Our data suggested that TET1 specifically regulates SIRT1-dependent deacetylation of PGC-1α and FOXO1, but not HIF-1α (*Figure 4*). Recently, AMPK-mediated TET2 phosphorylation at Ser99 was shown to be a molecular switch that controls a pivotal step in the AMPK-TET2-5hmC axis signaling pathway to regulate glucose synthesis (*Wu et al., 2018*). Although here it is not sure whether AMPK directly regulates phosphorylation of TET1, our findings suggest the notion that TET1 plays an important role in glucose metabolism through AMPK-dependent signaling by directly interacting with and activating SIRT1, supporting the notion that TET1 loss-of-function led to increased energy expenditure and protection from diet-induced obesity, insulin resistance, and glucose tolerance (*Damal Villivalam et al., 2020*). The finding that there was no alteration in the methylation level of the identified gene promoters, such as *Aldh1a3*, *Aldoa*, *Slc2a4*, *G6pc*, *Ppargc1a*, and *Ppargc1b*, in Tet1 knockout mESCs by analyzing the published data from *Dawlaty et al., 2011*; *Figure 8—figure supplement 1A*, and the 5hMC binding of the promoters of *PPARGC1A*, *G6PC*, and *SLC2A4* had also no change in TET1-depleted cells by hMeDIP-qPCR assays (*Figure 8—figure supplement 1B*). These data suggest that the regulation in the hydroxymethylation of these genes possibly does not affect their expression in Tet1 deficiency cells.

Previous studies show that TET1 can form a complex with transcriptional factors in several tissues or during different biological processes, such as the TET1/PRC1 complex in ESCs or the TET1/SIN3A/hMOF complex during DNA damage and repair (*Chandru et al., 2018*; *Zhu et al., 2018*; *Wu et al., 2011*; *Zhong et al., 2017*). Our findings identify the transcriptional regulation complex TET1/SIRT1 by verifying that the C-terminal of TET1 physically interacts with SIRT1 and that TET1 regulates SIRT1 deacetylase activity in vitro and in vivo (*Figure 3*). Although previous studies demonstrate that SIRT2 and SIRT6 also contribute to deacetylation of PGC-1α and FOXO1, thus regulating hepatic

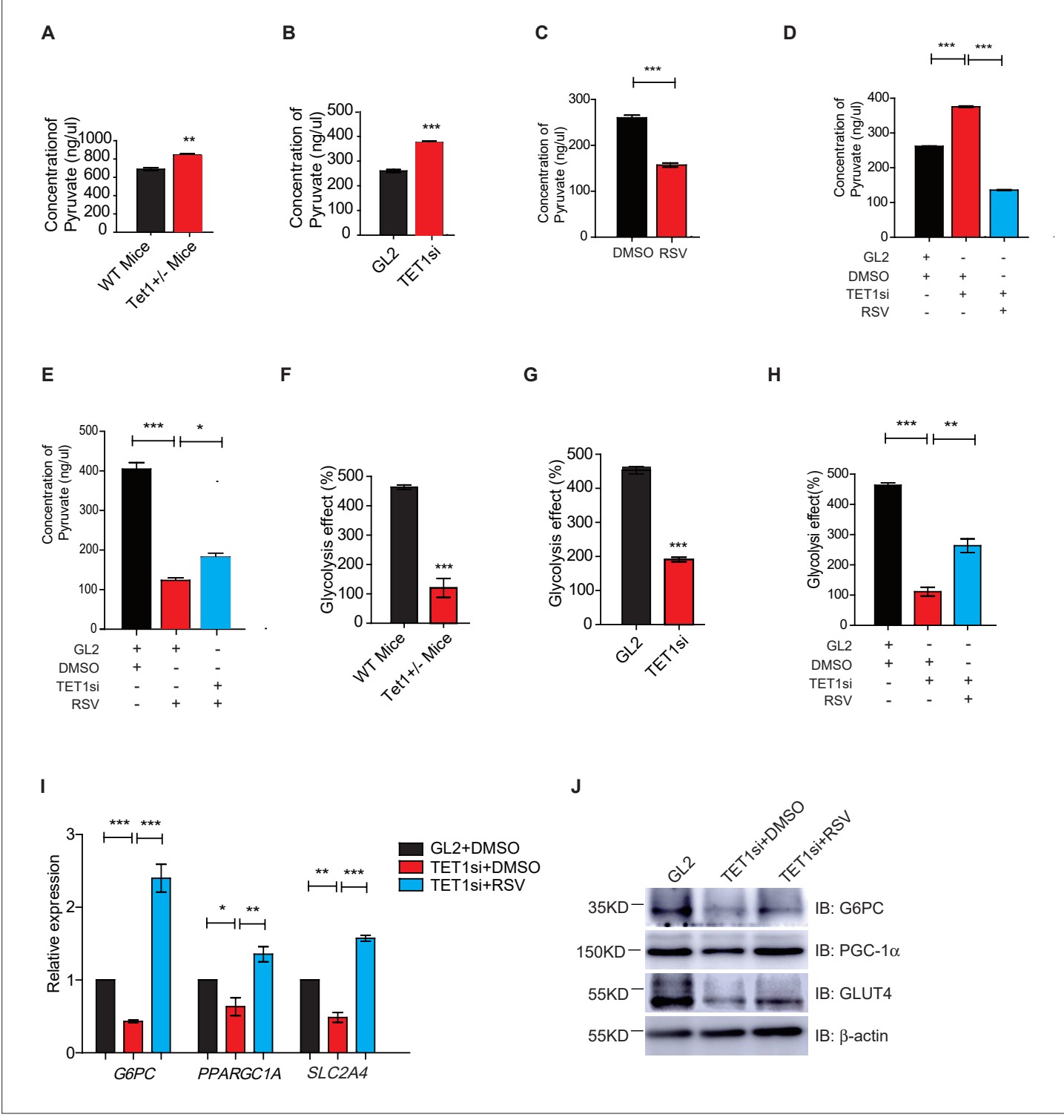

**Figure 5.** Silent information regulator T1 (SIRT1) activator resveratrol (RSV) rescues the ten-eleven translocation methylcytosine dioxygenase 1 (Tet1)-depleted cellular defect. (**A**) Comparison of the pyruvate level in liver cells between *Tet1*[+/-] mice with wild-type (WT) mice (**p < 0.01). (**B**) Comparison of the pyruvate level in HepG2 cells between TET1-depleted cells and the control (***p < 0.001). (**C**) Comparison of the pyruvate level in RSV-treated and DMSO-treated HepG2 cells (***p < 0.001). (**D**) Comparison of the pyruvate level among control HepG2 cells, TET1-depleted HepG2 cells, and TET1-depleted HepG2 cells after RSV treatment (***p < 0.001). (**E**) Comparison of the pyruvate level among control HepG2 cells, RSV-treated HepG2 cells, and TET1-depleted HepG2 cells after RSV treatment (*p < 0.05, ***p < 0.001). (**F** and **G**) Comparison of glycolysis effect in *Tet1*[+/-] mice liver tissue and the control, or in TET1-depleted HepG2 cells and the control (***p < 0.001). (**H**) Comparison of glycolysis effect among control HepG2 cells, RSV-treated

*Figure 5 continued on next page*

*Figure 5 continued*

HepG2 cells, and TET1-depleted HepG2 cells after RSV treatment (**p < 0.01, ***p < 0.001). (**I**) Comparison of the transcriptional expression among control HepG2 cells, TET1-depleted HepG2 cells, and TET1-depleted HepG2 cells after RSV treatment (*p < 0.05, **p < 0.01, ***p < 0.001). (**J**) Western blot analysis of the protein expression of *PPARGC1A*, *G6PC*, and *SLC2A4* among control HepG2 cells, TET1-depleted HepG2 cells, and TET1-depleted HepG2 cells after RSV treatment.

The online version of this article includes the following source data and figure supplement(s) for figure 5:

**Source data 1.** Supplementary Original Data - *Figure 5J*.

**Figure supplement 1.** The transcriptional expression of *G6PC*, *PPARGC1A,* and *SLC2A4* are regulated via silent information regulator T1 (SIRT1) deacetylase activity.

**Figure supplement 1—source data 1.** Supplementary Original Data *Figure 5—figure supplement 1C*.

**Figure supplement 2.** Ten-eleven translocation methylcytosine dioxygenase 1 (TET1) regulates gluconeogenic gene expression via silent information regulator T1 (SIRT1) activity.

**Figure supplement 3.** Resveratrol (RSV) could rescue the decrease of glucose level, the defects in glucose and insulin tolerance, and gluconeogenesis of *Tet1*$^{+/-}$ mice.

gluconeogenesis, our results show that the C-terminal of TET1 specifically interacts with SIRT1 but not SIRT2 and SIRT6. Interestingly, studies also suggest that the region within the TET1 catalytic domain (TET1-CD) is responsible for binding to O-linked N-acetylglucosamine (O-GlcNAc) transferase (*Zhong et al., 2017*; *Hrit et al., 2018*; *Bauer et al., 2015*; *Delatte, 2014*; *Zhang et al., 2014*; *Vella et al., 2013*), hMOF (*Zhong et al., 2017*), and HIF-2α (*Tsai et al., 2014*). Rinf (i.e., CXXC5) also interacts with the C-terminal of both TET1 and TET2 to form a transcriptional regulatory complex (*Ravichandran et al., 2019*). These findings support the notion that TET1, especially the TET1-CD, functions as a coactivator to specifically interact with SIRT1 to regulate downstream events. Structural analysis of catalytic domains can provide more details about TET functionality.

Dynamic regulation of transcriptional factor PGC-1α/FOXO1 localization, which is dependent on acetylation, is an important manner in the transcriptional regulation of gluconeogenesis (*Anderson et al., 2008*; *Matsuzaki et al., 2005*; *Qiang et al., 2010*). Our findings suggest that TET1 also regulates the translocalization of PGC-1α/FOXO1, which is dependent on SIRT1 acetylation but not HIF-1α. Collectively, these studies suggest that TET1 specifically activates these transcription factors through SIRT1 activity. Given the established relationship between TET1 and the translocalization of PGC-1α/FOXO1, it is likely that the effect of PGC-1α acetylation is because of inhibition of SIRT1. However, we cannot completely rule out the possibility that other sirtuins play a role in translocalization of PGC-1α/FOXO1 in TET1-depleted cells.

Additionally, TET1 did not significantly affect the protein levels and chromatin localization of AMPK and p-AMPK (*Figure 6*). Glucagon activates AMPK activity to regulate glucose metabolism during fasting, and abnormal glucagon secretion is associated with type 1 and type 2 diabetes (*Leclerc et al., 2011*). Our findings reveal that TET1 is also required for the activation of transcriptional expression of *G6PC*, *SLC2A4*, and *PPARGC1A* induced by glucagon. These results suggest that the AMPK-TET1-SIRT1 axis is required for glucose metabolism induced by AMPK activation. Notably, AICAR and metformin activate the PGC-1α pathway and alter mitochondrial protein levels as well as cellular stress pathways in a strong cell-type- and dose-dependent manner (*Figure 6E*). A previous study also showed that incubation of HepG2 cells with higher concentrations of AICAR (1 mM) increased fatty acid oxidation and gluconeogenesis, while lower concentrations (0.05 and 0.1 mM) of AICAR decreased fatty acid oxidation and gluconeogenesis (*Lee et al., 2006*).

*Tet1*$^{+/-}$ mice exhibited a series of functional defects in glucose metabolism, such as significant decreases in blood glucose, increased glucose tolerance, and insulin sensitivity, suggesting that Tet1 functions as an activator in gluconeogenesis. Tet1 depletion blocks the decrease of blood glucose in cells or mice induced by metformin or RSV, suggesting that Tet1 plays an important role in AMPK-SIRT1-dependent glucose metabolism. Another contribution of this study was that we first determined the important role of Tet1 in glucose metabolism. TET1 also promotes fatty acid oxidation and inhibits NAFLD progression by hydroxymethylation of PPARα promoter (*Wang et al., 2020*).

Overall, our study indicates that activation of the AMPK-TET1-SIRT1 axis leads to the translocalization of PGC-1α/FOXO1, which is dependent on the acetylation level and promotes gluconeogenic gene programs in the liver, which has a profound impact on glucose metabolism in fasting.

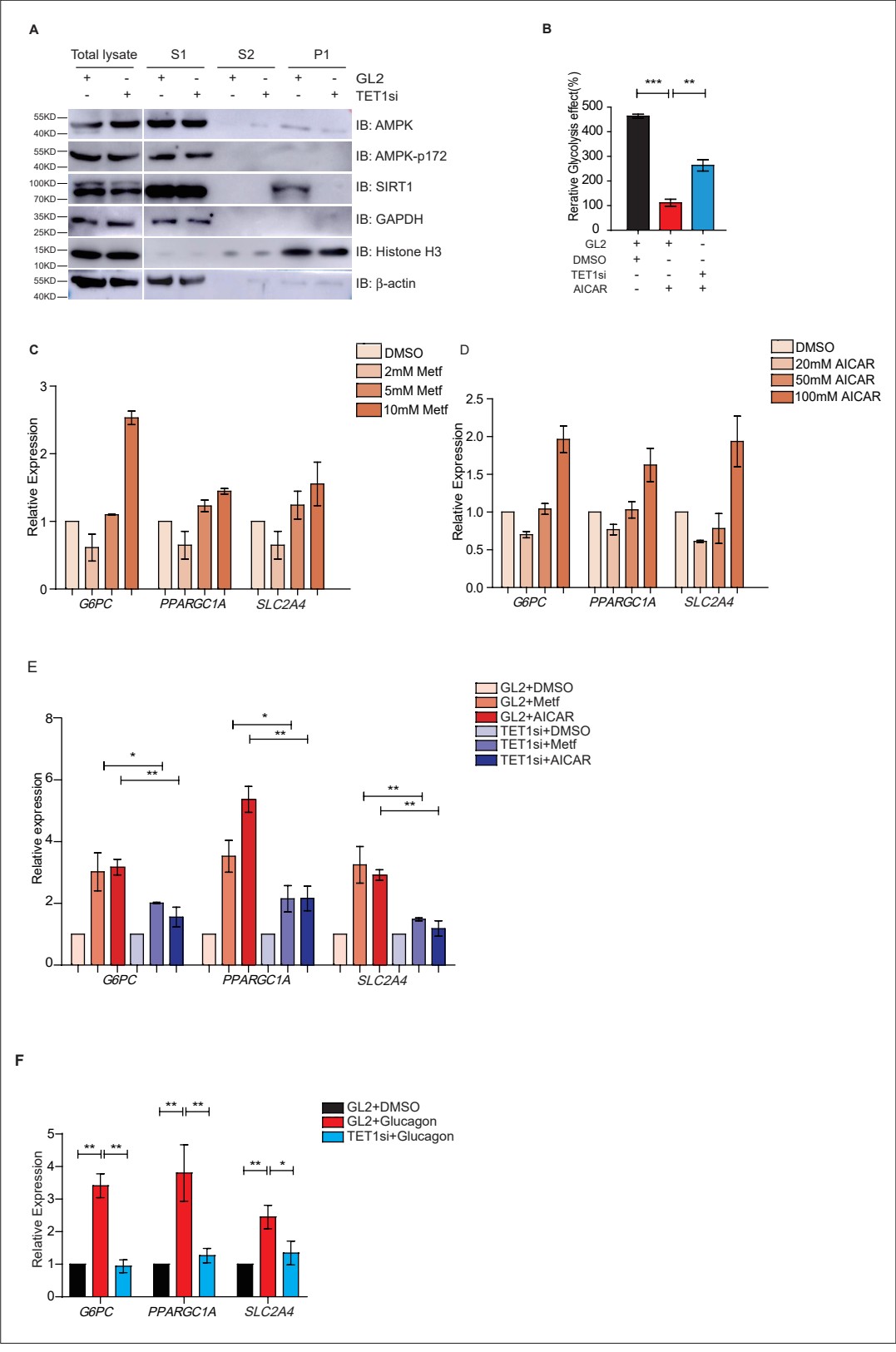

**Figure 6.** Ten-eleven translocation methylcytosine dioxygenase 1 (TET1) depletion blocks gluconeogenic gene expression activation induced by adenosine 5'-monophosphate-activated protein kinase (AMPK). (**A**) Western blot analysis of the fractions in TET1 knockdown HepG2 cells using indicated antibodies. (**B**) Comparison of the effect on glycolysis among control HepG2 cells, 5-aminoimidazole-4-carboxamide ribonucleoside (AICAR)-

*Figure 6 continued on next page*

*Figure 6 continued*

treated HepG2 cells, and TET1-depleted HepG2 cells after AICAR treatment (*p < 0.05, ***p < 0.001). (**C**) qRT-PCR analysis of the transcriptional expression of the selected genes in gluconeogenesis, including *PPARGC1A*, *G6PC*, and *SLC2A4*, with a dosage treatment of metformin in HepG2 cells for 8 hr (*p < 0.05, **p < 0.01, ***p < 0.001). (**D**) qRT-PCR analysis of the transcriptional expression of the selected genes in gluconeogenesis, including *PPARGC1A*, *G6PC*, and *SLC2A4*, with a dosage treatment of AICAR in HepG2 cells for 8 hr (*p < 0.05, **p < 0.01). (**E**) Comparison of the transcriptional expression among control HepG2 cells, metformin or AICAR-treated HepG2 cells, and TET1-depleted HepG2 cells after metformin or AICAR treatment, (*p < 0.05, **p < 0.01). (**F**) Comparison of the transcriptional expression among control HepG2 cells, glucagon (0.1 nM)-treated HepG2 cells, and TET1-depleted HepG2 cells after glucagon treatment. (*p < 0.05, **p < 0.01).

The online version of this article includes the following figure supplement(s) for figure 6:

**Figure supplement 1.** Silent information regulator T1 (SIRT1) cannot affect the localization and activation of adenosine 5'-monophosphate-activated protein kinase (AMPK).

Further studies are required to clarify the signaling axis effects and analyze the protein structure of TET1 targeting SIRT1, providing more precise targets for the clinical therapy to type 2 diabetes or metabolism-related diseases.

# Materials and methods

## Key resources table

| Reagent type (species) or resource | Designation | Source or reference | Identifiers | Additional information |
|---|---|---|---|---|
| Gene (*Homo sapiens* [*human*]) | *TET1* | GenBank | RC218608 | |
| Gene (*Homo sapiens* [*human*]) | *SIRT1* | GenBank | NM_012238 | |
| Gene (*Homo sapiens* [*human*]) | *SIN3A* | GenBank | NM_001145357 | cDNA were purchased from Origene Corp. |
| Cell line (*Homo sapiens* [*human*]) | HepG2 cell line | ATCC | HB-8065 | |
| Cell line (*Homo sapiens* [*human*]) | LO2 cell line | Cellosaurus | CVCL_6926 | |
| Cell line (*Homo sapiens* [*human*]) | HEK293T cell line | ATCC | CRL-11268 | |
| Transfected construct (human) | TET1 shRNA #1; shRNA #2 | lentivirus vector pLKO_SHC201 | Cat# SHC201 | Sequence: shRNA#1: CCCAGAAGATTT AGAATTGAT shRNA#2: CCTCCAGTCTT AATAAGGTTA |
| Transfected construct (human) | siRNA to TET1 (SMARTpool) | Dharmacon/Thermo Fisher Scientific | Cat# D-014635 | |
| Transfected construct (human) | siRNA: non-target in control (GL2 Duplex) | Dharmacon/Thermo Fisher Scientific | 1022070 | |
| Sequence-based reagent | *G6PC*. S | This paper | RT-PCR primers | GCAGGTGTATAC TACGTGATGGT |
| Sequence-based reagent | *G6PC*. A | This paper | RT-PCR primers | GACATTCAAGC ACCGAAATCTG |
| Sequence-based reagent | *PPARGC1A*. S | This paper | RT-PCR primers | TGACTGGCGTC ATTCAGGAG |
| Sequence-based reagent | *PPARGC1A*. A | This paper | RT-PCR primers | CCAGAGCAG CACACTCGAT |
| Sequence-based reagent | *SLC2A4*. S | This paper | RT-PCR primers | GGGAAGGAAA AGGGCTATGCTG |

*Continued on next page*

*Continued*

| Reagent type (species) or resource | Designation | Source or reference | Identifiers | Additional information |
|---|---|---|---|---|
| Sequence-based reagent | *SLC2A4*. A | This paper | RT-PCR primers | CAATGAGGAAC CGTCCAAGAATG |
| Sequence-based reagent | *TET1*. S | This paper | RT-PCR primers | GGAATGGAAG CCAAGATCAA |
| Sequence-based reagent | *TET1*. A | This paper | RT-PCR primers | ACTCCCTAAG GTTGGCAGTG |
| Sequence-based reagent | *PCK1*.S | This paper | RT-PCR primers | ATCCCCAAAAC AGGCCTCAG |
| Sequence-based reagent | *PCK1*. A | This paper | RT-PCR primers | ACGTACATGGT GCGACCTTT |
| Sequence-based reagent | *ACTB*. S | This paper | RT-PCR primers | TACTGCCCTG GCTCCTAGCA |
| Sequence-based reagent | *ACTB*. A | This paper | RT-PCR primers | GCCAGGATAGA GCCACCAATC |
| Antibody | Anti-SIRT1 (Rabbit polyclonal) | Millipore | Cat# 05–1243 | WB (1:2000) |
| Antibody | Anti-TET1 (Rabbit polyclonal) | Millipore | Cat# 09–872 | WB (1:1000) |
| Antibody | Anti-SIN3A (Rabbit polyclonal) | Abcam | Cat# ab3479 | WB (1:2000) |
| Antibody | Anti-AMPK (Rabbit polyclonal) | Abcam | Cat# ab3759 | WB (1:2000) |
| Antibody | Anti-AMPK T172 (Rabbit polyclonal) | Cell Signaling | Cat# 2535 | WB (1:2000) |
| Antibody | Anti-p53 (Rabbit polyclonal) | Abcam | Cat# ab26 | WB (1:2000) |
| Antibody | Anti-p53 acetyl 382 (Rabbit polyclonal) | Abcam | Cat# ab75754 | WB (1:2000) |
| Antibody | Anti-FOXO1 (Rabbit polyclonal) | Abcam | Cat# ab52857 | WB (1:2000) IF (1:500) |
| Antibody | Anti-PGC-1$\alpha$ (Rabbit polyclonal) | Abcam | Cat# ab54481 | WB (1:2000) IF (1:500) |
| Antibody | Anti-Histone H3 (Mouse monoclonal) | Abcam | Cat# ab1791 | WB (1:4000) |
| Antibody | Anti-lysine acetylated antibody (Rabbit polyclonal) | Cell Signaling | Cat# 9441 | WB (1:2000) |
| Antibody | Anti-LaminB (Rabbit polyclonal) | Santa Cruz | Cat# sc-6216 | WB (1:4000) |
| Antibody | Anti-GAPDH (Mouse monoclonal) | Santa Cruz | Cat# sc-32233 | WB (1:4000) |
| Antibody | Anti-5hMC (Rabbit polyclonal) | Active Motif | Cat# AB_2630381 | IP 4 μg/1 mg total lysate |
| Chemical compound, drug | Metformin | Sigma Aldrich | Cat# PHR108 | |
| Chemical compound, drug | AICAR | Sigma Aldrich | Cat# A9978 | |

*Continued*

| Reagent type (species) or resource | Designation | Source or reference | Identifiers | Additional information |
|---|---|---|---|---|
| Chemical compound, drug | RSV | Sigma Aldrich | Cat# R5010 | |
| Chemical compound, drug | EX527 | Sigma Aldrich | Cat# E7034 | |
| Chemical compound, drug | SRT2104 | SellectChem | Cat# S7792 | |
| Commercial assay or kit | Pyruvate Assay Kit | Sigma Aldrich | Cat# MAK071 | |
| Commercial assay or kit | Glucose (Go) Assay Kit | Sigma Aldrich | Cat# GAGO20 | |
| Commercial assay or kit | SIRT1 Activity Assay Kit | Abcam | Cat# ab156065 | |
| Software, algorithm | SPSS; Image J | SPSS; Image J | RRID:SCR_002865; RRID:SCR_003070 | |
| Strain, strain background (*Escherichia coli*) | BL21(DE3) | Sigma Aldrich | CMC0016 | |
| Other | DAPI stain | Invitrogen | D1306 | (1 µg/ml) |
| Other | Lipo3000 | Invitrogen | Cat# L3000001 | |
| Other | Tet1 Knockout mice | Jackson | Cat# 017358 | |

## Mice and animal experiments

*Tet1*$^{+/-}$ mice were obtained from the Jackson Laboratory (Bar Harbor, ME, USA, Cat# 017358). For genotyping of *Tet1*$^{+/-}$ mice, the forward primer, AACTGATTCCCTTCGTGCAG, and the reverse primer, TTAAAGCATGGGTGGGAGTC, were used. The expected band size for homozygote mutant was 650 bp, 850 bp for the WT strain, and 650 bp and 850 bp double bands for the heterozygote strain. Male WT and *Tet1* heterozygous mice (7–8 weeks old) were maintained on a C57BL/6 J background. Mice were randomly divided into different groups and ad libitum fed with Chow or HFD (60 kcal% fat, D12492). For RSV treatment, mice were treated with 20 mg/kg/day of RSV by intragastric administration or free feeding HFD supplemented with 0.4 % RSV (4 g/kg diet). For metformin treatment, mice were administrated with 150 mg/kg/day metformin in drinking water for a specific period. Groups that were treated with ddH$_2$O served as control. Animal experiments were conducted in accordance with an approved protocol by the Institutional Animal Care and Ethics Committee of Xiamen University and Nanchang University.

## Cell culture, plasmids, antibodies, primers, siRNA oligonucleotides, and chemicals

All the cells were cultured in DMEM media (Hyclone, Logan, UT, USA), supplemented with 10 % fetal bovine serum (Hyclone, Logan, UT, USA) and penicillin-streptomycin (Invitrogen, Carlsbad, CA, USA). The cell lines were negative for mycoplasma contamination, not in the list of commonly misidentified cell lines maintained by the International Cell Line Authentication Committee, and authenticated by STR profiling. RNA interference (RNAi) experiments were performed using Dharmacon siGENOME SMARTpool siRNA duplexes (Thermo Fisher Scientific, Carlsbad, CA, USA) against TET1 (Cat#D-014635). GL2 is the siRNA control targeting to luciferase. Cells were transfected with an siRNA complex at a final concentration of 20 nM using RNAiMAX transfection Lipo3000 reagent (Invitrogen, Carlsbad, CA, USA, Cat#L3000001). TET1 cDNA was purchased from Origene (RC218608). cDNA for SIRT1 (NM_012238) and SIN3A (NM_001145357) were generated from the cDNA library and confirmed by DNA sequencing, and then subcloned into pcDNA3.0 vector or pGEX-4T vector, followed by sequencing validation: Metformin (Sigma Aldrich, Louis, MO, USA, Cat#PHR1084), AICAR (Sigma Aldrich, Louis, MO, USA, Cat#A9978), RSV (Sigma Aldrich,

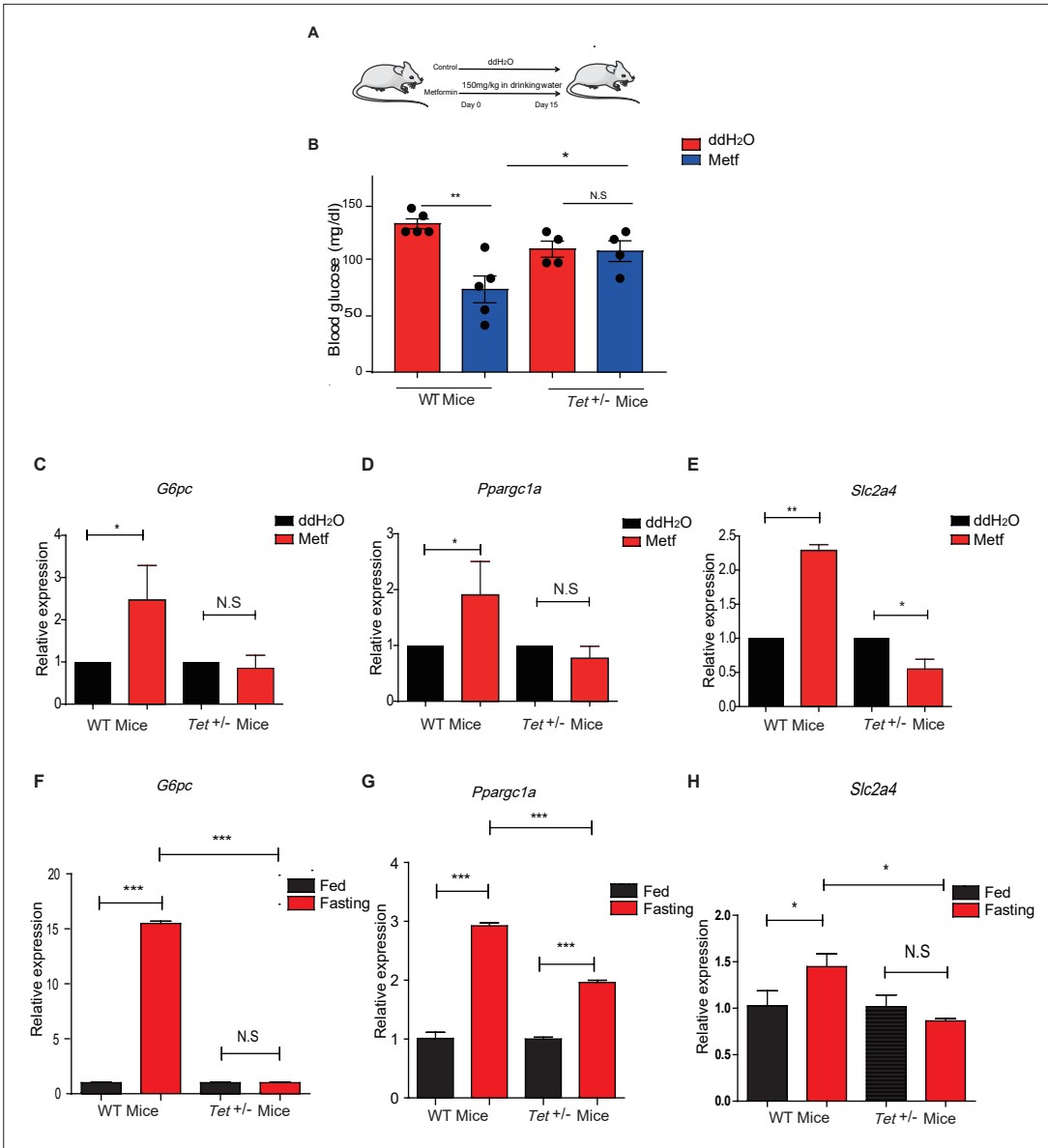

**Figure 7.** Metformin or fasting affects ten-eleven translocation methylcytosine dioxygenase 1 (Tet1) regulation of gluconeogenesis-related transcriptional expression. (**A**) The schedule of metformin treatment in $Tet1^{+/-}$ male mice and wild-type (WT) male mice. (**B**) The glucose levels of $Tet1^{+/-}$ male mice and WT male mice before and after metformin treatment ($*p < 0.05$, $**p < 0.01$, N = 5). (C, D, and E) Comparison of the transcriptional expression of $G6pc$, $Ppargc1a$, and $Slc2a4$ genes between $Tet1^{+/-}$ mice with WT mice before and after metformin treatment ($*p < 0.05$, $**p < 0.01$). Normalization of $Tet1^{+/-}$ mice and WT mice as the control. (F, G, and H) Comparison of the transcriptional expression of $G6pc$, $Ppargc1a$, and $Slc2a4$ genes via qRT-PCR analysis between $Tet1^{+/-}$ mice with WT mice before and after the fasting for 24 hr ($*p < 0.05$, $***p < 0.001$, NS means $p > 0.05$). Normalization of $Tet1^{+/-}$ mice and WT mice as the control.

The online version of this article includes the following figure supplement(s) for figure 7:

**Figure supplement 1.** The body weight of $Tet1^{+/-}$ mice has no alternation compared with wild-type (WT) mice with or without metformin treatment.

USA, Louis, MO, Cat#R5010), EX527 (Sigma Aldrich, Louis, MO, USA, Cat#E7034), SRT2104 (SellectChem, Radnor, PA, USA, Cat# S7792). Primary antibodies used as following: SIRT1 (Millipore, Louis, MO, 05–1243), TET1 (Millipore, Louis, MO, 09–872), SIN3A (Abcam,Waltham, MA, USA, ab3479), AMPK (Abcam, Waltham, MA, USA, ab3759), AMPK T172 (Cell Signaling, #2535), p53 (Abcam, Waltham, MA, USA, ab26), p53 acetyl 382 (Abcam, Waltham, MA, USA, ab75754), p53

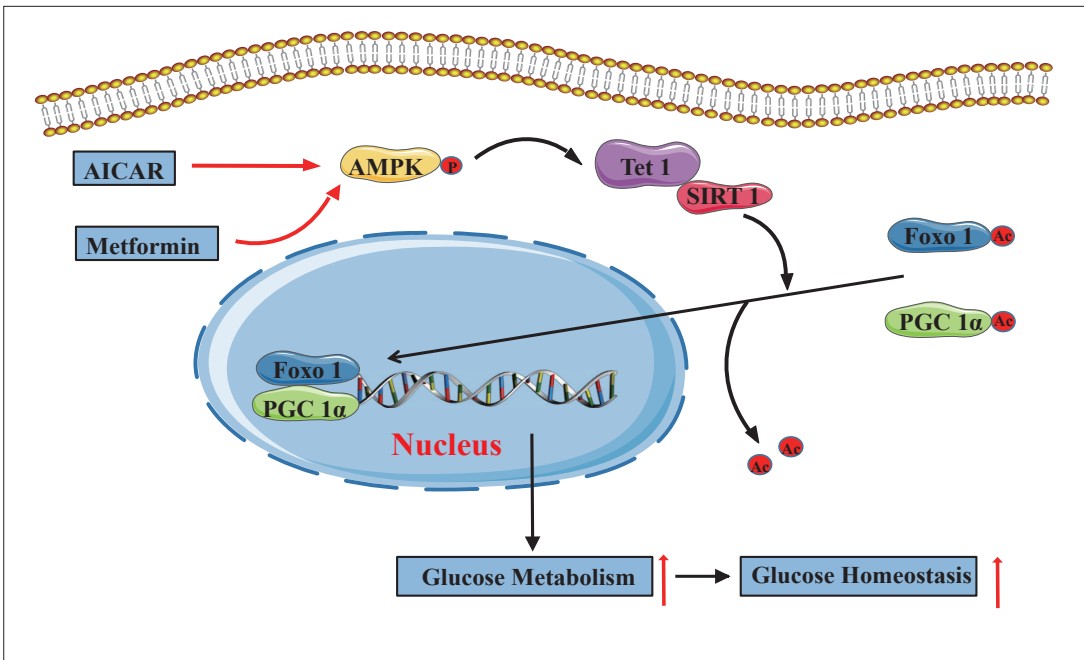

**Figure 8.** The adenosine 5'-monophosphate-activated protein kinase (AMPK)/ten-eleven translocation methylcytosine dioxygenase 1 (TET1)/silent information regulator T1 (SIRT1) axis in hepatic glucose metabolism. Model showing the mechanism of the AMPK-TET1-SIRT1 axis regulating the deacetylation-dependent translocalization of PGC-1α and FOXO1 from the cytoplasm to the nucleus, which contributes to glucose homeostasis via hepatic glucose metabolism.

The online version of this article includes the following figure supplement(s) for figure 8:

**Figure supplement 1.** 5-Hydroxymethylcytosine (5hMC) in the promoters of some gluconeogenic genes has no change in in ten-eleven translocation methylcytosine dioxygenase 1 (TET1) shRNA-inhibited LO2 cells.

---

acetyl 120 (Abcam, Waltham, MA, USA, ab78316), FOXO1 (Abcam, Waltham, MA, USA, ab52857), PGC-1α (Abcam, Waltham, MA, USA, ab54481), histone H3(Abcam, Waltham, MA, USA, ab1791), lysine acetylated antibody (Cell Signaling, Danvers, MA, USA, #9441), LaminB (Santa Cruz, Santa, CA, USA, sc-6216), GAPDH (Santa Cruz, Santa, CA, USA, sc-32233), 5hMC (Active Motif, Carlsbad, CA,USA, AB_2630381), His-tag (Sigma Aldrich, Louis, MO, USA,, SAB1306085), GST-tag (Abgent, Shanghai, China, AM1011a), Flag-tag (Sigma Aldrich, Louis, MO, USA,, F7425), and β-actin (Sigma Aldrich, Louis, MO, USA, A1978). Second antibodies used as follows: anti-mouse IgG peroxidase labeled antibodies (Sigma Aldrich, Louis, MO, USA, A4416), anti-rabbit IgG peroxidase labeled antibodies (Sigma Aldrich, Louis, MO, A6154), anti-rabbit IgG Alexa Fluor 594 fluorescence labeled antibody (Jackson, West Grove, PE, USA, 111-585-003), and anti-mouse IgG Alexa Fluor 488 fluorescence labeled antibody (Jackson, West Grove, PE, USA, 715-546-150). TET1 shRNA1: CCCA GAAGATTTAGAATTGAT; TET1 shRNA 2: CCTCCAGTCTTAATAAGGTTA; primers were shown in *Supplementary file 1*.

## Tissue RNA isolation and qRT-PCR

Mice were anesthetized with chloral hydrate, and ~50 mg liver tissue from each mouse was collected in a 1.5 ml tube, quick-frozen in liquid nitrogen, and the tissue sample was ground with a pestle immediately. RNA samples were isolated using a Trizol RNA Extraction Kit (Invitrogen, Carlsbad, CA, USA, Cat# 15596018) following the manufacturer's protocol. cDNA was reverse-transcribed from total RNA extracted from cells using the First Strand Reverse Transcription Kit (Fermentas, Carlsbad, CA, USA, Cat# K1691). Four microliters (~80 ng) of the cDNA were used for qRT-PCR using specific primers of mouse genes. qRT-PCR was performed on an ABI 7500 Real-time PCR system using the SYBR Green probe (Roche, Germany, Cat# 04707516001).

## GST pull-down assay

GST-tagged proteins, His-SIN3A and His-SIRT1, were induced using IPTG in BL21 bacterial cells and were further purified following the manual of Glutathione-Sepharose 4B (GE Health, Boston, MA, USA, Cat# 17-0756-01) and His-tag Purification Resin (Roche, Germany, Cat# 08778850001). GST or GST-fusion protein was incubated with the whole cell lysates or His-SIRT1 as indicated in the results and prepared with Glutathione-Sepharose beads at 4 °C overnight. After centrifuging at 1500 $g$ for 10 min, the pellets were washed four times with 100 bed volumes of NETN buffer (0.5% NP-40, 0.1 mM EDTA, 20 mM Tris-HCl, pH 7.4, 300 mM NaCl). The pellets were eluted by heating at 100 °C for 10 min in SDS-PAGE loading buffer.

## Immunofluorescence staining

Cells grown on a glass coverslip were fixed in 4 % paraformaldehyde/PBS for 15 min at RT and then permeabilized with PBS containing 0.1 % Triton X-100 (PBST) for 15 min at RT. After blocking in PBS with 5 % BSA for 30 min, cells were incubated with anti-PGC-1α or anti-FOXO1 in PBS/5 % BSA for 1 hr at RT. Cells were washed three times with PBST and then incubated with secondary antibodies, and then Alexa Fluor 546 (Goat anti-mouse IgG, 1:500) and/or Alexa Fluor 488 (Goat anti-rat IgG, 1:200) in PBS/5 % BSA at RT for 1 hr. After three washes in PBST, cells were stained with DAPI for 2 min and mounted with nail polish. Fluorescence in cells was monitored using a laser scanning microscope (Carl Zeiss 880, Germany) for confocal microscopy and a Zeiss software package for image acquisition.

## Chromatin fractionation

Cultured cells were harvested and lysated in CEBN buffer 10 mM HEPES pH 7.8, 10 mM KCl, 1.5 mM MgCl$_2$, 0.34 M sucrose, 10 % glycerol, 0.2% NP-40, 1 × protease inhibitor cocktail (Roche, Germany, Cat# 11836153001), 1 × phosphatase Inhibitor cocktail (Roche, Germany, Cat# 11697498001) and $N$-ethylmaleimide. After centrifuging at 1500 $g$ for 10 min at 4 °C, the samples were washed in ice-cold CEB buffer once (CEBN buffer without NP-40), and then the pellets were further lysated in soluble nuclear buffer (3 mM EDTA, 0.2 mM EGTA, inhibitors as described above). The resultant suspensions were centrifuged at 2000 $g$ for 10 min at 4 °C and washed in ice-cold soluble nuclear buffer once. Sequentially, the pellets were added to salt buffers with high NaCl concentrations (50 mM Tris-HCl pH 8.0, 0.05% NP-40, 0.45 M NaCl, 1 × protease inhibitor cocktail).

## Pyruvate assay kit, glucose (Go) assay kit and SIRT1 activity assay kit

Pyruvate, glucose detection, and SIRT1 activity experiments were performed using a pyruvate assay kit, glucose assay kit, and SIRT1 activity assay kit, respectively, which were purchased from Sigma Aldrich Co. Ltd (Louis, MO, USA, Cat# MAK071 and Cat# GAGO20) and Abcam Co. Ltd (Waltham, MA, USA, Cat# ab156065). Pyruvate, glucose level, and SIRT1 activity were determined according to the corresponding manufacturer's instructions.

## GTT, ITT, and PTT

PTT was performed following 12 hr overnight fasting. Each animal received an i.p. injection of 2 g/kg pyruvate (Sigma Aldrich, Louis, MO, USA, Cat#P2256) in sterile saline according to the amount of 0.1 ml/10 g. GTT was performed following 8  and 10 hr overnight fasting, respectively. Each animal received an i.p. injection of 2 g/kg dextrose (Aladdin Industrial Corp.Shanghai, China, Cat# G116302) in sterile saline according to the amount of 0.1 ml/10 g. ITT was performed following daytime fasting from 9 am to 3 pm. Each animal received an i.p. injection of 0.75 U/kg insulin in sterile saline according to the amount of 0.1 ml/10 g. Blood glucose concentrations were measured using a GA-3 glucometer via tail vein bleeding at the indicated time points (0, 30, 60, and 120 min) after injection. For all animal experiments, age-matched animals were used. Insulin resistance (HOMA-IR) was calculated using the following equation: HOMA-IR = (fasting insulin (ng/ml) × fasting plasma glucose (mg/dl)/405).

siRNA transfection, RNA isolation, and qRT-PCR siRNA transfections were carried out with Lipofectamine 2000 (Invitrogen, Carlsbad, CA, USA), and all indicated plasmids transfections were performed with polyethylenimine (Bender, Austria) according to the manufacturer's instructions. Total RNAs from all indicated cells were isolated with Trizol reagent (Invitrogen, Carlsbad, CA, USA). cDNA was reverse-transcribed from total RNA extracted from HEK293T cells using the First Strand Reverse Transcription

Kit (Fermentas, Carlsbad, CA, USA,). qRT-PCR was performed on an ABI 7300 Real-time PCR system using the SYBR Green probe (Takara, Japan).

### Immunoblotting

Cells were washed with ice-cold PBS, scraped, and transferred into 1.5 ml microcentrifuge tubes that were spun for 1 min at maximum speed. The pellet of cells was lysated in 100 mM HEPES containing 200 mM NaCl, 10 % glycerol, 2 mM $Na_4P_2O_7$, 2 mM DTT, 1 mM EDTA, 1 mM benzamidine, 0.1 mM $Na_3VO_4$, and protease inhibitor cocktail at pH 7.4. After heated at 100 °C for 10 min, the cell lysates were centrifuged at 12,000 $g$ for 10 min. The resulted supernatant containing the equivalent amounts of protein were loaded to 10–12% SDS-PAGE for gel-electrophoresis and were transferred onto nitrocellulose membrane (Bio-Rad, USA) for 3 hr on ice. Blotting membranes were incubated with blocking solution 5 % nonfat dried milk powder dissolved in TBST buffer (pH 7.5, 10 mM Tris-HCl, 150 mM NaCl, and 0.1 % Tween 20) for 1 hr at RT. After washed three times, the membrane was incubated with targeted antibodies in TBST buffer overnight at 4 °C. After several washes with TBST buffer, the membranes were incubated for 1 hr with horseradish peroxidase-linked secondary antibody (1:5000). The membranes were then processed with enhanced chemiluminescence Western blotting detection reagents (Pierce, Louis, MO, USA). Exposure of the membranes was covered by films.

### Co-immunoprecipitation

Cells were washed in PBS and lysed in 800 µl lysis buffer (50 mM Tris-HCl pH 7.4, 200 mM NaCl, 1 mM EDTA, 1% NP-40, 0.02 % sodium azide, 0.1 % SDS, 1 % PMSF, and protease inhibitor cocktail) for 30 min. Then, the lysates were centrifuged at 12,000 $g$ for 15 min, and the supernatant was incubated for 1 hr at 4 °C with prepared 50 % protein A-sepharose beads (Roche, USA) and followed by centrifugation to remove proteins that adhered nonspecifically to the beads. The purified supernatant was mixed with protein A-sepharose beads and antibodies overnight. The resultant immune bound proteins were isolated by 2000 $g$ centrifugation. After washing three times with washing buffer (50 mM Tris-HCl pH 7.4, 200 mM NaCl, 1 mM EDTA pH 8.0, 1% NP-40, 0.02 % sodium azide, 0.1 % SDS, 1 % PMSF), the immune bound proteins were eluted by heating at 100 °C for 10 min in SDS-PAGE loading buffer.

### Statistical Analysis

Statistical analysis was performed by using the Student's t-test, one-way ANOVA, and nonparametric test. $p < 0.05$ was considered significantly different.

### Acknowledgements

We are grateful to Prof. Jian Zhang, Prof. Gonghua Hu for equipment supply, Dr Jinfang Zhang for critically evaluating the manuscript, and the other members in our lab for their invaluable assistance. We also thank LetPub (https://www.letpub.com/) for its linguistic assistance and scientific consultation during the preparation of this manuscript.

### Additional information

#### Funding

| Funder | Grant reference number | Author |
| --- | --- | --- |
| National Natural Science Foundation of China | 81760160 | Jianing Zhong |
| Startup Fund for Scholars of Gannan Medical University | QD201605 | Jianing Zhong |
| Innovative Team of Gannan Medical University | TD201708 | Jianing Zhong |

| Funder | Grant reference number | Author |
| --- | --- | --- |
| The Open Project of Key Laboratory of Prevention and Treatment of Cardiovascular and Cerebrovascular Diseases, Ministry of Education | XN201807 | Jianing Zhong |
| JiangXi Provincial Natural Science Foundation | 20202BAB206086 | Jianing Zhong |
| JiangXi Provincial Natural Science Foundation | 20171ACB21001 | Chunbo Zhang |
| JiangXi Provincial Natural Science Foundation | 20171BCB23029 | Chunbo Zhang |
| National Natural Science Foundation of China | 82072302 | Jianing Zhong |

The funders had no role in study design, data collection and interpretation, or the decision to submit the work for publication.

## Author contributions

Chunbo Zhang, Conceptualization, Funding acquisition, Investigation, Methodology, Project administration, Validation, Writing – original draft, Writing – review and editing; Tianyu Zhong, Yuanyuan Li, Investigation, Methodology; Xianfeng Li, Data curation, Formal analysis; Xiaopeng Yuan, Linlin Liu, Weilin Wu, Jing Wu, Ye Wu, Rui Liang, Xinhua Xie, Data curation, Methodology; Chuanchuan Kang, Yuwen Liu, Zhonghong Lai, Jianbo Xiao, Zhixian Tang, Riqun Jin, Yan Wang, Methodology; Yongwei Xiao, Investigation, Methodology, Validation; Jin Zhang, Data curation, Validation; Jian Li, Investigation, Validation; Qian Liu, Formal analysis, Visualization; Zhongsheng Sun, Formal analysis, Investigation, Validation; Jianing Zhong, Conceptualization, Funding acquisition, Project administration, Resources, Supervision, Writing – original draft, Writing – review and editing

## Author ORCIDs

Chunbo Zhang ⓘ http://orcid.org/0000-0001-9618-9415
Jianing Zhong ⓘ http://orcid.org/0000-0002-2781-3437

## Ethics

Animal experiments were conducted in accordance with an approved protocol by the Institutional Animal Care and Ethics Committee of Xiamen University and Nanchang University.

## Decision letter and Author response

Decision letter https://doi.org/10.7554/eLife.70672.sa1
Author response https://doi.org/10.7554/eLife.70672.sa2

# Additional files

## Supplementary files

- Transparent reporting form
- Supplementary file 1. List of primers.

## Data availability

All data generated or analysed during this study are included in the manuscript and supporting files.

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
