## [Editor Report]

The authors identify the DNA demethylase Tet1 as being a critical regulator of hepatic carbohydrate metabolism. Moreover, the authors discover these effects are mediated through a novel non-canonical function of Tet1 independent of altering DNA hydroxymethylation.

---

## [Decision Letter]

**Decision letter after peer review:**

Thank you for submitting your article "The hepatic AMPK-TET1-SIRT1 axis regulates glucose homeostasis" for consideration by *eLife*. Your article has been reviewed by 3 peer reviewers, one of whom is a member of our Board of Reviewing Editors, and the evaluation has been overseen by Mone Zaidi as the Senior Editor. The following individual involved in review of your submission has agreed to reveal their identity: Mingming Gao (Reviewer #2).

Essential revisions:

1) The concerns over HepG2 cells, a cancer line, and their physiological relevance to hepatic metabolism was raised as a major concern. Primary murine hepatocytes should be utilized at least for seminal studies central to the conclusions of the manuscript. Speicifically, does TET1 knockdown alter gluconeogenic gene expression profiles in a more physiologically relevant model system?

2) 5hMC of target loci needs to be evaluated. Evaluating 5mC from publicly available datasets generated in TET1 deficient mESCs is insufficient to draw conclusions that TET1 is acting through a non-canonical pathway independent of methylation patterns.

*Reviewer #1:*

Zhang et al., demonstrate a role of TET1 in the regulation of hepatic gluconeogenesis. Through the use of in vivo and in vitro studies the authors identified mechanistically an interaction between the DNA demethylase TET1 and the deacetylase SIRT1. Furthermore, the authors demonstrate that this interaction is essential for the translocation of the transcription factors FOXO1 and PGC1a to induce the transcription of gluconeogenic genes. The major strengths of this manuscript include the detailed mechanistic nature of the experiments. The authors utilize both functional and mechanistic studies. The in vivo functional studies performed by Zhang et al. show in rodents TET1 deficiency improves glucose tolerance. Furthermore, the authors employed high fat diet in rodents as a model of glucose intolerance to show that TET1 deficiency improves glucose metabolism. in vitro, the authors employed classical biochemical assays showing a physical interaction between SIRT1 and TET1. Furthermore, through truncation mutant studies the authors identified the C-terminal region of TET1 as mediating this interaction. Moreover, through pharmacological studies the authors demonstrate a role for AMPK in the regulation of SIRT1/TET1 interaction. Overall, this is a well conducted study with potential for offering new therapeutic avenues for the treatment of glucose metabolic disorders.

This is a well conducted study by the authors. There are many strengths of this study (see public review). However, there are some issues that should be addressed. First, the authors tend to use drugs/compounds or call out protein names without any explanation to what the drug is used for or the rationale for assaying that pathway (e.g. SIN3a in line 178, EX527 in line 181 and GL2 in line 203).

A major concern are the authors conclusion that methylation is not the mechanism for their observed findings. TET1 is classically viewed as a DNA demethylase through the conversion of 5mC to 5hmC to eventually lead to the removal of methyl groups on DNA. The authors provide previously published data showing track browsers for DNA methylation of target genes in TET1 KO in mESC without any subsequent changes in 5mC. This is a premature claim by the authors. First, mESCs are not a good representation of a gluconeogenic cell line. Furthermore, the assessment of 5mC falls short. What happens to 5hmC at these loci? To claim a non-canonical role of TET1 in the regulation of gluconeogenic gene expression in the absence of assessing 5hmC at these loci is insufficient. A recommendation would be to assess 5hmC at select loci in either Tet1+/- livers or from Tet1 siRNA in hepG2 cells to functionally demonstrate whether methylation patterns are affecting gene expression profiles in this context.

*Reviewer #2:*

The manuscript titled "the hepatic AMPK-TET1-SIRT1 axis regulates glucose homeostasis" provides comprehensive evidence suggesting that TET1 plays a pivotal role in regulating hepatic glucose metabolism by regulating SIRT1-mediated protein deacetylation. The tolerance tests data of WT vs heterozygous group are impressive though no body weight change, suggesting that TET1 is important for regulating hepatic gluconeogenesis. This is further supported by gene expression analysis data showing heterozygous mice had upregulated glycolysis genes while downregulated gluconeogenesis genes. Detailed biochemical and biophysical analysis suggests that TET1 binds to SIRT1 to increase its deacetylase activity, leading to nucleus translocation of PGC1-α and FOXO1. This is further strengthened by additional evidence showing TET1 functions downstream of AMPK though detailed mechanism remains open. Importantly, many of these interactions were further verified using agonists and antagonists to stimulate or rescue the signal transduction and functional outcomes.

Please see below my comments/suggestions to the authors.

1. Figure 1, it would be nice to show protein and/or gene expression changes of TET1 in WT vs heterozygous mice. I would assume that compared to WT, the heterozygous had a downregulated expression of TET1.

2. In Figure 2, the authors showed that heterozygous mice had downregulated expression of Glut4. The physiologic significance of this downregulation remains unclear to me, considering the fact that Glut4 is mainly expressed in the muscle and adipose tissues.

3. Figure 2B, the authors also provided IF data from KO mice. Is this from embryos? I ask this question because the authors mentioned in the first paragraph of the results part that homozygous KO led to embryonic lethality.

4. Figure 3F, the EX527 SIRT1 inhibitor was used at a high concentration. Potential off target effects of this tool compound cannot be ruled out.

5. Figures 2-6, HepG2 line was used as an in vitro model. Considering HepG2 is a cancer cell line in which glucose metabolism and signal transduction could be distinct from normal hepatocytes, these data could be strengthened by adding other lines or using primary cells for some key experiments.

6. Figure 5. RSV is not a selective agonist of SIRT1. GSK2245840/SRT2104 is a selective activator of SIRT1 currently in phase 2 trial.

7. Figure 6F, glucagon was used at superphysiological levels. The physiological range of glucagon is usually below 100 pmol/L, even under fasting conditions.

8. It would be nice if the authors could discuss little bit more about how to target TET1 for treating metabolic disorders.

*Reviewer #3:*

1. The study mainly demonstrated how TET1 regulates glucose metabolism, especially glucose tolerance and insulin resistance, while most of the mechanistic studies were performed in a hepatic cancer cell line, HepG2. Given huge metabolic differences between normal cells and cancer cells, the reviewer strongly recommends the authors to repeat the mechanistic studies in normal human and mouse hepatic cells, ideally using primary cells.

2. The authors only used single siRNA to knock down TET1. To avoid potential off-target effect, two independent siRNAs is required.

3. TET1 depletion does not affect AMPK activation, while the authors conclude that TET1 regulates genes like G6PC, PPARGC1A and GLUT4 expression through AMPK. Please clarify.

4. In Figure 3C, co-IP followed by western blot for SIN3A is needs. In Figure 3D, full length TET1 control is needed. In Figure 3G, full length TET1 and FL1+FL2 are needed for controls.

5. PPARGC1A, G6PC and GLUT4 are the key readout for findings, thus only RT-qPRC is not sufficient. Western blot is required to measure the expression levels of the genes, such as Figure 4F, 4G, 5I, 5J, 6C-6G.

---

## [Author Response]

Essential revisions:1) The concerns over HepG2 cells, a cancer line, and their physiological relevance to hepatic metabolism was raised as a major concern. Primary murine hepatocytes should be utilized at least for seminal studies central to the conclusions of the manuscript. Speicifically, does TET1 knockdown alter gluconeogenic gene expression profiles in a more physiologically relevant model system?

Thank you for your suggestion. Here, we further employed human normal liver cells (LO2 cell line) as a more physiologically relevant model system for addressing the question whether TET1 regulates the transcriptional expression of gluconeogenic genes. Therefore, we repeated some key experiments in LO2 cells to confirm the conclusion that Tet depletion can downregulate the transcriptional expression of PPARGC1A, G6PC and GLUT4 genes using Tet1 shRNA-knockdown cells. To keep the main text's integrity and consistency in this manuscript, we arrange these results as the supplementary data in figure 2-supplement 1. We revised the main text as the following:

“The results showed that TET1 knockdown can lead to the increase of the selected genes in glycolysis and decrease of the genes in gluconeogenesis. We further verified the transcriptional expression (figure 2-supplement 1A) and protein level (figure 2-supplement 1B) of PPARGC1A, G6PC, and GLUT4 genes in TET1 knockdown LO2 cells using two shRNA respectively, suggesting TET1 can play important roles in the expression regulation of PPARGC1A, G6PC, and GLUT4 genes.”

2) 5hMC of target loci needs to be evaluated. Evaluating 5mC from publicly available datasets generated in TET1 deficient mESCs is insufficient to draw conclusions that TET1 is acting through a non-canonical pathway independent of methylation patterns.

I agree with you and thank you for your suggestion. The level of 5hMC is relatively low in cancer cells. We also think Tet1 regulates the transcriptional expression of downstream genes via activating the SIRT1 activity, not excluding the possibility that the mechanism is independent of TET1’s demethylation activity. Here, we further adopted the hMeDIP-qPCR using 5hMC specific antibody to evaluate the target loci of 5hMC in LO2 cells (human normal liver cells), supporting the hypothesis that TET1 is acting through a non-canonical pathway independent of methylation patterns. And we revised the main text in the Discussion section as the following:

“The finding that there was no alteration in the methylation level of the identified gene promoters, such as Aldh1a3, Aldoa, Glu4, G6pc, Ppargc1a, and Ppargc1b, in TET1 knockout mESCs by analyzing the published data from Meelad M Dawlaty et al. ^(14)^ (figure 8-supplement 1A), and the 5hMC binding of the promoters of PPARGC1A, G6PC, and GLUT4 had also no change in TET1-depleted cells by hMeDIP-qPCR assays (figure 8-supplement 1B). These data suggest that the regulation in the hydroxymethylation of these genes does not possibly affect their expression in Tet1 deficiency cells.”

Reviewer #1:Zhang et al., demonstrate a role of TET1 in the regulation of hepatic gluconeogenesis. Through the use of in vivo and in vitro studies the authors identified mechanistically an interaction between the DNA demethylase TET1 and the deacetylase SIRT1. Furthermore, the authors demonstrate that this interaction is essential for the translocation of the transcription factors FOXO1 and PGC1a to induce the transcription of gluconeogenic genes. The major strengths of this manuscript include the detailed mechanistic nature of the experiments. The authors utilize both functional and mechanistic studies. The in vivo functional studies performed by Zhang et al. show in rodents TET1 deficiency improves glucose tolerance. Furthermore, the authors employed high fat diet in rodents as a model of glucose intolerance to show that TET1 deficiency improves glucose metabolism. in vitro, the authors employed classical biochemical assays showing a physical interaction between SIRT1 and TET1. Furthermore, through truncation mutant studies the authors identified the C-terminal region of TET1 as mediating this interaction. Moreover, through pharmacological studies the authors demonstrate a role for AMPK in the regulation of SIRT1/TET1 interaction. Overall, this is a well conducted study with potential for offering new therapeutic avenues for the treatment of glucose metabolic disorders.This is a well conducted study by the authors. There are many strengths of this study (see public review). However, there are some issues that should be addressed. First, the authors tend to use drugs/compounds or call out protein names without any explanation to what the drug is used for or the rationale for assaying that pathway (e.g. SIN3a in line 178, EX527 in line 181 and GL2 in line 203).

Thank you for your suggestion. According to your suggestion, we have revised the main text in manuscript via adding the explanation about drugs and protein names including SIN3a, EX527, GL2, and SRT2104 in the picture annotation and Materials section.

A major concern are the authors conclusion that methylation is not the mechanism for their observed findings. TET1 is classically viewed as a DNA demethylase through the conversion of 5mC to 5hmC to eventually lead to the removal of methyl groups on DNA. The authors provide previously published data showing track browsers for DNA methylation of target genes in TET1 KO in mESC without any subsequent changes in 5mC. This is a premature claim by the authors. First, mESCs are not a good representation of a gluconeogenic cell line. Furthermore, the assessment of 5mC falls short. What happens to 5hmC at these loci? To claim a non-canonical role of TET1 in the regulation of gluconeogenic gene expression in the absence of assessing 5hmC at these loci is insufficient. A recommendation would be to assess 5hmC at select loci in either Tet1+/- livers or from Tet1 siRNA in hepG2 cells to functionally demonstrate whether methylation patterns are affecting gene expression profiles in this context.

It is a good point, and I agree with you. The level of 5hMC is relatively low in cancer cells. We also think Tet1 regulates the transcriptional expression of downstream genes via activating the SIRT1 activity, not excluding the possibility that the mechanism is independent of TET1’s demethylation activity. Here, we further adopted the hMeDIP-qPCR using 5hMC specific antibody to evaluate the target loci of 5hMC in LO2 cell lines, supporting the hypothesis that TET1 is acting through a non-canonical pathway independent of methylation patterns. And we modified the main text in Discussion section as the following:

“The finding that there was no alteration in the methylation level of the identified gene promoters, such as Aldh1a3, Aldoa, Glu4, G6pc, Ppargc1a, and Ppargc1b, in TET1 knockout mESCs by analyzing the published data from Meelad M Dawlaty et al. ^(14)^ (Figure 8-supplement 1A), and the 5hMC binding of the promoters of PPARGC1A, G6PC, and GLUT4 had also no change in TET1-depleted cells by hMeDIP-qPCR assays (Figure 8-supplement 1B). These data suggest that the regulation in the hydroxymethylation of these genes does not affect their expression in Tet1 deficiency cells.”

Reviewer #2:The manuscript titled "the hepatic AMPK-TET1-SIRT1 axis regulates glucose homeostasis" provides comprehensive evidence suggesting that TET1 plays a pivotal role in regulating hepatic glucose metabolism by regulating SIRT1-mediated protein deacetylation. The tolerance tests data of WT vs heterozygous group are impressive though no body weight change, suggesting that TET1 is important for regulating hepatic gluconeogenesis. This is further supported by gene expression analysis data showing heterozygous mice had upregulated glycolysis genes while downregulated gluconeogenesis genes. Detailed biochemical and biophysical analysis suggests that TET1 binds to SIRT1 to increase its deacetylase activity, leading to nucleus translocation of PGC1-α and FOXO1. This is further strengthened by additional evidence showing TET1 functions downstream of AMPK though detailed mechanism remains open. Importantly, many of these interactions were further verified using agonists and antagonists to stimulate or rescue the signal transduction and functional outcomes.Please see below my comments/suggestions to the authors.1. Figure 1, it would be nice to show protein and/or gene expression changes of TET1 in WT vs heterozygous mice. I would assume that compared to WT, the heterozygous had a downregulated expression of TET1.

Thank you for your suggestion. According to your suggestion, we further check the gene expression changes of TET1 in live tissue using RT-PCR. The results showed that the transcriptional expression of Tet1 in heterozygous mice is reduced by half compared to that in WT. We added the figure 1-supplement 1A and revised the main text as the following:

“Since Tet1 homozygous mice (Tet1^-/-^) exhibit embryonic lethality ^(13,14)^, we used Tet1 heterozygous mice (Tet1^+/-^) and verified its transcriptional expression of Tet1 (Figure 1-supplement 1A).”

2. In Figure 2, the authors showed that heterozygous mice had downregulated expression of Glut4. The physiologic significance of this downregulation remains unclear to me, considering the fact that Glut4 is mainly expressed in the muscle and adipose tissues.

It is a good point and I agree with you. GLUT4 is mainly expressed in the muscle and adipose tissues, fasting and insulin cause reduced GLUT4 expression in both white and brown adipose tissues. We also observed the protein level of GLUT4 in different tissue in Tet1 heterozygous mice. It is reported that GLUT4 is an important downstream gene in dependent on the transcriptional factors, such as PGC1a and FOXO1. In our study, we verified the transcriptional expression of GLUT4 in fasting and insulin-induced mice models. Therefore, GLUT4, as a glucose transporter, is significantly downregulated via Tet1-dependent manner which could contribute to regulate glucose transport and metabolism in the fasting condition.

3. Figure 2B, the authors also provided IF data from KO mice. Is this from embryos? I ask this question because the authors mentioned in the first paragraph of the results part that homozygous KO led to embryonic lethality.

Thank you for your question. We performed the IF assays using HepG2 cells, heterozygous or homozygous KO mice tissue. In previous studies, homozygous Tet1 KO mice led to embryonic lethality (Hao Wu et al., 2011; Shinpei Yamaguchi et al., 2012). We also confirmed this phenotype in our mice. However, we also found that a very small number of mice can survive, which exhibit defects in development and smaller body weight. This similar phenotype was also observed in previous publication. We think that there may be a compensatory effect in the TET family. Therefore, we chose and performed the most experiments in TET1 heterozygous mice to accurately reflect the regulatory role of TET in glucose metabolism.

4. Figure 3F, the EX527 SIRT1 inhibitor was used at a high concentration. Potential off target effects of this tool compound cannot be ruled out.

Thank you for your suggestion. Potential off target effects of EX527 need to be considered. So far, EX527 is still the most effective selective inhibitor of SIRT1.EX 527 effectively inhibits the activity of SIRT1 deacetylase with IC50 of 38nM, which is concentration-dependent, while inhibiting SIRT2 and SIRT3, the effect is much lower, respectively. Considering this possibility, we performed the transcriptional expression of downstream genes in EX527 dose dependent manner. We treated with cells using EX527 reference concentration, which specifically inhibits SIRT1 activity.

5. Figures 2-6, HepG2 line was used as an in vitro model. Considering HepG2 is a cancer cell line in which glucose metabolism and signal transduction could be distinct from normal hepatocytes, these data could be strengthened by adding other lines or using primary cells for some key experiments.

Thank you for your suggestion. Here, we further employed another hepatocyte cell line (LO2) as a relevant model for addressing the question whether TET1 regulates the transcriptional expression of gluconeogenic genes. We repeated some key experiments in LO2 cell line to confirm the conclusion that Tet depletion can downregulate the transcriptional expression of PPARGC1A, G6PC and GLUT4 genes using Tet1 shRNA knockdown. To keep the main text's integrity and consistency in this manuscript, we arrange these results as the supplementary data in Figure 2-supplement 1. We revised the main text as the following:

“The results showed that TET1 knockdown can lead to the increase of the selected genes in glycolysis and decrease of the genes in gluconeogenesis. We further verified the transcriptional expression (Figure 2-supplement 1A) and protein level (Figure 2-supplement 1B) of PPARGC1A, G6PC, and GLUT4 genes in TET1 knockdown LO2 cells using two shRNA respectively, suggesting TET1 can play important roles in the expression regulation of PPARGC1A, G6PC, and GLUT4 genes.”

6. Figure 5. RSV is not a selective agonist of SIRT1. GSK2245840/SRT2104 is a selective activator of SIRT1 currently in phase 2 trial.

Thank you for your suggestion. We have performed the same experiment using SRT2104 instead of RSV to confirm the results. The results showed that SRT2104 can rescue the transcriptional expression in TET1 depleted cells. These data were arranged in Figure 5-supplement 1 and the main text was revised as the following:

“RSV and SIRT1 specific activator SRT2104 can increase the protein expression of G6PC, PPARGC1A, and GLUT4 in cells (Figure 5-supplement 1C). Interestingly, RSV or SRT2104 treatment rescued the decreased gluconeogenic transcriptional in TET1-depleted cells (Figure 5I, J). TET1 depletion blocked the increase of gluconeogenic gene expression—including G6PC, PPARGC1A, and GLUT4—in RSV-treated cells (Figure 5-supplement 1D) or SIRT2104-treated cells (Figure 5-supplement 1E).”

7. Figure 6F, glucagon was used at superphysiological levels. The physiological range of glucagon is usually below 100 pmol/L, even under fasting conditions.

Thank you for your suggestion. According to your suggestion, we further repeated these assays using a lower concentration of glucagon. The concentration of glucagon used in cell models often stimulates gene transcription at higher concentrations. This concentration is indeed relatively high for simulated physiology. In this experiment, we tried to prove the molecular mechanism that the transcriptional expression of glucagon activated cells depends on TET1, which has certain limitations. Accordingly, we have deleted figure 6F and repeated and modified figure 6G with 0.1nM glucagon. The data showed the similar results.

8. It would be nice if the authors could discuss little bit more about how to target TET1 for treating metabolic disorders.

Thank you for your suggestion. We do not have enough evidence to support the idea how to target TET1 for treating metabolic disorders. We will pay more effort to address this question in future. In Discussion section, we added the sentence as the following:

“Further studies are required to clarify the signaling axis effects and analyze the protein structure of TET1 targeting SIRT1, providing more precise targets for the clinical therapy to type 2 diabetes or metabolism related diseases. ”

Reviewer #3:1. The study mainly demonstrated how TET1 regulates glucose metabolism, especially glucose tolerance and insulin resistance, while most of the mechanistic studies were performed in a hepatic cancer cell line, HepG2. Given huge metabolic differences between normal cells and cancer cells, the reviewer strongly recommends the authors to repeat the mechanistic studies in normal human and mouse hepatic cells, ideally using primary cells.

Thank you for your suggestion. We repeated the some key experiments in human normal liver cell (LO2) to confirm the conclusion that Tet depletion can downregulate the transcriptional expression and protein expression of PPARGC1A, G6PC and GLUT4 genes using two Tet1 shRNA knockdown. For keeping the integrity and consistency of the main text in this manuscript, we arrange these results in Figure 2-supplement 1 and revised the main text as the following:

“The results showed that TET1 knockdown can lead to the increase of the selected genes in glycolysis and decrease of the genes in gluconeogenesis. We further verified the transcriptional expression (Figure 2-supplement 1A) and protein level (Figure 2-supplement 1B) of PPARGC1A, G6PC, and GLUT4 genes in TET1 knockdown LO2 cells using two shRNA respectively, suggesting TET1 can play important roles in the expression regulation of PPARGC1A, G6PC, and GLUT4 genes.”

2. The authors only used single siRNA to knock down TET1. To avoid potential off-target effect, two independent siRNAs is required.

Thank you for your suggestion. We employed the siRNA pool purchased from Dharmacon which contains four siRNA. It was proved that this siRNA has a good inhibitory effect and high efficiency in our previous published article (Jianing Zhong et al., 2017). Here, we also designed two shRNA to repeat some important experiments in LO2 cell line. The results showed that TET1 depletion using shRNA could downregulate the transcriptional expression of PPARGC1A, G6PC and GLUT4 genes.

3. TET1 depletion does not affect AMPK activation, while the authors conclude that TET1 regulates genes like G6PC, PPARGC1A and GLUT4 expression through AMPK. Please clarify.

Thank you for your question. AMP-activated protein kinase (AMPK) plays a major role in regulating glucose metabolism as an early signal sensor. AMPK activity is stimulated by phosphorylation of threonine 172 (Thr 172) (Robin Willows et al., 2017). Metformin can regulate glucose metabolism in AMPK dependent manner. Our data showed that AMPK activated the downstream genes in cells and glucose metabolism in mice depending on TET1. Accordingly, the upstream and downstream relationship between AMPK and TET1 needs to be addressed. Our data indicated that TET1 regulated genes like G6PC, PPARGC1A, and GLUT4 expression through AMPK, but not affected AMPK activation, suggesting that AMPK is an upstream factor of TET1 as the earlier signal sensor to regulate these genes. Therefore, we conclude that TET1 regulates genes like G6PC, PPARGC1A, and GLUT4 expression through AMPK.

4. In Figure 3C, co-IP followed by western blot for SIN3A is needs. In Figure 3D, full length TET1 control is needed. In Figure 3G, full length TET1 and FL1+FL2 are needed for controls.

Thank you for your suggestion. We consider the full length TET1 and FL1+FL2 fragment are important, and also construct the plasmid of FL1+FL2 (1-1590aa). Unfortunately, the full length and FL1+FL2 sequence is too long, resulting in a very small amount of expressed protein, which is not suitable for IP experiments and pull-down assays in our cellular system. In fact, we did the IP assay by endogenous protein using TET1 specific antibody to support the interaction between TET1 and SIRT1. The constructs of FL1, FL2 and FL3 were used in our published article (Jianing Zhong et al., 2017). The results showed that FL3 played important roles in interacting with several proteins, such as hMOF, SIN3A, and HIF. SIN3A as a cofactor of SIRT1 has proved to interact with TET1, which is a positive control. Accordingly, it is hard to add the full length and FL1+FL2 plasmids for IP and Pull down assays.

5. PPARGC1A, G6PC and GLUT4 are the key readout for findings, thus only RT-qPRC is not sufficient. Western blot is required to measure the expression levels of the genes, such as Figure 4F, 4G, 5I, 5J, 6C-6G.

Thank you for your suggestion. Our key point is that the transcriptional expression of these genes are downregulated in Tet1 deficiency. According to your suggestion, we have measured the protein expression of these genes in Tet1shRNA knockdown cells.